# Forcing For Varying Boundary Layer Stability Across Antarctica

Mckenzie J. Dice [1,2,3]

John J. Cassano [1,2,3]

Gina C. Jozef [1,2,3]

[1] Department of Atmospheric and Oceanic Sciences, University of Colorado Boulder

[2] Cooperative Institute for Research in Environmental Sciences, University of Colorado Boulder

[3] National Snow and Ice Data Center, University of Colorado Boulder

**Key points**

- A consistent decrease in downwelling longwave radiation as stability increases points to radiative forcing as an important distinguishing factor in changing stability across the Antarctic continent and throughout all seasons, except summer.
- Between the three weakest stability regimes (near-neutral, very shallow mixed, and weak stability), a unique pattern emerges, and likely defines the very shallow mixed regime where wind speeds are always weaker in comparison to the near-neutral and weak stability regimes.
- Reduced downwelling longwave radiation in combination with stronger near-surface wind speed is associated with stability regimes with enhanced stability above a layer of weaker near-surface stability.

Corresponding Author: Mckenzie Dice, mckenzie.dice@colorado.edu

## Abstract

The relative importance of changes in radiative forcing (downwelling longwave radiation) and mechanical mixing (20 m wind speed) in controlling boundary layer stability annually and seasonally at five study sites across the Antarctica continent is presented. From near-neutral to extremely strong near-surface stability, radiative forcing decreases with increasing stability, as expected, and is shown to be a major driving force behind variations in near-surface stability at all five sites. Mechanical mixing usually decreases with increasing near-surface stability for regimes with weak to extremely strong stability. For the cases where near-neutral, very shallow mixed, and weak stability occur, the wind speed in the very shallow mixed case is usually weaker compared to the near-neutral and weak stability cases while radiative forcing is largest for the near-neutral cases. This finding is an important distinguishing factor for the unique case where a very shallow mixed layer is present, indicating that weaker mechanical mixing in this case is likely responsible for the shallower boundary layer that defines the very shallow mixed stability regime. For cases with enhanced stability above a layer of weaker near-surface stability, lower downwelling longwave radiation promotes the persistence of the stronger stability aloft, while stronger near-surface winds act to maintain weaker stability immediately near the surface, resulting in this two-layer boundary layer stability regime.

## 1 Introduction

The atmospheric boundary layer is the lowest part of the atmosphere where the surface of the earth and overlying atmosphere interact, for example, exchanging heat and moisture. Boundary layer stability varies based largely on the surface energy budget and mechanical mixing driven by wind shear. Increased downwelling longwave radiation, in the presence of cloud cover, or solar radiation reduces boundary layer stability, while clear skies, with less downwelling longwave radiation, and long periods of darkness, especially in the polar regions, allows for the formation of strong near-surface temperature inversions (King and Turner 1997; Cassano et al., 2016). Increased near surface wind speed, and thus wind shear, can also reduce stability by generating turbulence and mixing down warmer air from aloft. In contrast, weak winds and reduced wind shear and mixing allow for stronger near-surface stability (Hudson and Brandt, 2005; Dice and Cassano, 2022). Here, we will use the findings from Dice et al. (2023), which described the range of boundary layer stability present at two continental interior and three coastal sites in Antarctica (Figure 1), to determine how radiative forcing and mechanical mixing vary across this range of boundary layer stability regimes, and how these mechanisms vary seasonally and across the continent.

Previous boundary layer studies have widely documented radiative forcing and wind shear to be two main drivers of variations in static stability in the boundary layer (Hudson and Brandt, 2005; Stone and Kahl, 1991; King and Turner, 1997; etc.). In terms of radiative forcing, Cassano et al. (2016) found a strong seasonal cycle of inversion strength over the Ross Ice Shelf, approximately 100 km from McMurdo, with stronger inversion strength in the austral winter during polar night while solar radiation is zero, and weaker inversion strength in the austral summer during polar day when the sun is always above the horizon and solar heating is strongest. Dice and Cassano (2022) also found decreasing radiative flux with increasing stability at McMurdo. At Neumayer, Silva et al. (2022) noted strong temperature inversions, especially during the winter when solar radiation is low or zero during polar night. Hudson and Brandt (2005) found that inversion strength decreases with increasing radiative flux in the winter at South Pole and Dome Concordia ("Dome C"). This was also observed by Pietroni et al. (2013), who found the strongest surface-based temperature inversions at Dome C to occur with strong radiative cooling, which is at its maximum in the austral winter. Further, increased downwelling longwave radiation is usually associated with reduced near-surface stability in the Arctic (Solomon et al., 2023) and Antarctic (Stone and Kahl, 1991; Dice and Cassano, 2022).

When analyzing boundary layer stability in terms of variations in near-surface wind speed, Cassano et al. (2016) found that over the Ross Ice Shelf, the strength of inversions is related to the strength of the wind speed, with the strongest inversions occurring when the wind speed is less than 4 m s$^{-1}$, and the strength of the inversion rapidly decreases with increasing winds above 4 m s$^{-1}$. Dice and Cassano (2022) found the strongest inversions occurring with wind speeds less than 4.3 m s$^{-1}$ at McMurdo. Silva et al. (2022) investigated boundary layer stability at Neumayer, and found strong inversions were associated with low wind speeds. Hudson and Brandt (2005) found that, while it is generally expected that increasing wind speeds will reduce near-surface stability by mixing warmer air from aloft to the surface, the strongest stability conditions occur when wind speeds are 3 m s$^{-1}$ to 5 m s$^{-1}$ rather than calm at South Pole and Dome C, which was also noted by other studies in the coastal regions of Antarctica (Cassano et al., 2016). Results from Argentini et al. (2005) show that a wind speed of 4.5 m s$^{-1}$ is required to reduce stable conditions to well mixed conditions at Dome C and Pietroni et al. (2013) found the strongest surface-based temperature inversions at Dome C occur with weak winds.

In addition to radiative forcing and mechanical mixing, several other phenomena can alter the static stability in the boundary layer. For example, temperature advection or warm air from over open

water or cold air from over ice sheets or sea ice can quickly change near-surface stability conditions
especially at coastal locations such as McMurdo, Neumayer, and Syowa. Warm air advection over a cold
surface would result in increased near-surface stability (Stone and Kahl, 1991; Vignon et al., 2017;
Pietroni et al., 2013). It is also possible that cyclonic activity can alter near-surface boundary layer
stability, through changes in wind speed and cloud cover associated with the cyclone. At Neumayer,
cyclonic activity reaches a maximum in the fall and spring, during which temperature inversions are
rarely observed, whereas during non-cyclonic periods, temperature inversions are observed three times as
often compared to during the cyclonic periods (Silva et al., 2022). Additionally, katabatic flow from the
continental interior has been observed to impact conditions at Syowa and can flow from the plateau
located just above the South Pole as well. At Syowa, there is a high frequency of strong wind events
associated with katabatic activity in the fall and winter (Yamada and Hirasawa, 2018). The effects of
katabatic flow on boundary layer stability at Syowa is not well documented, but katabatic flow can either
result in the influx of cold air near the surface, resulting in strong temperature inversions, or increased
mechanical mixing can reduce near-surface stability (Vihma et al., 2011). At South Pole, clear-sky
conditions are associated with weak katabatic flow from the plateau, resulting in a persistent and strong
surface-based temperature inversion (Stone and Kahl, 1991).
The results presented below will assess differences in radiative forcing, as shown by downwelling
longwave radiation, and mechanical generation of turbulence, as shown by near-surface wind speed,
associated with varying boundary layer stability across the Antarctic continent (Section 3). The relative
importance of forcing mechanisms for the various regimes annually and seasonally and across the
continent will be explored in Section 4.

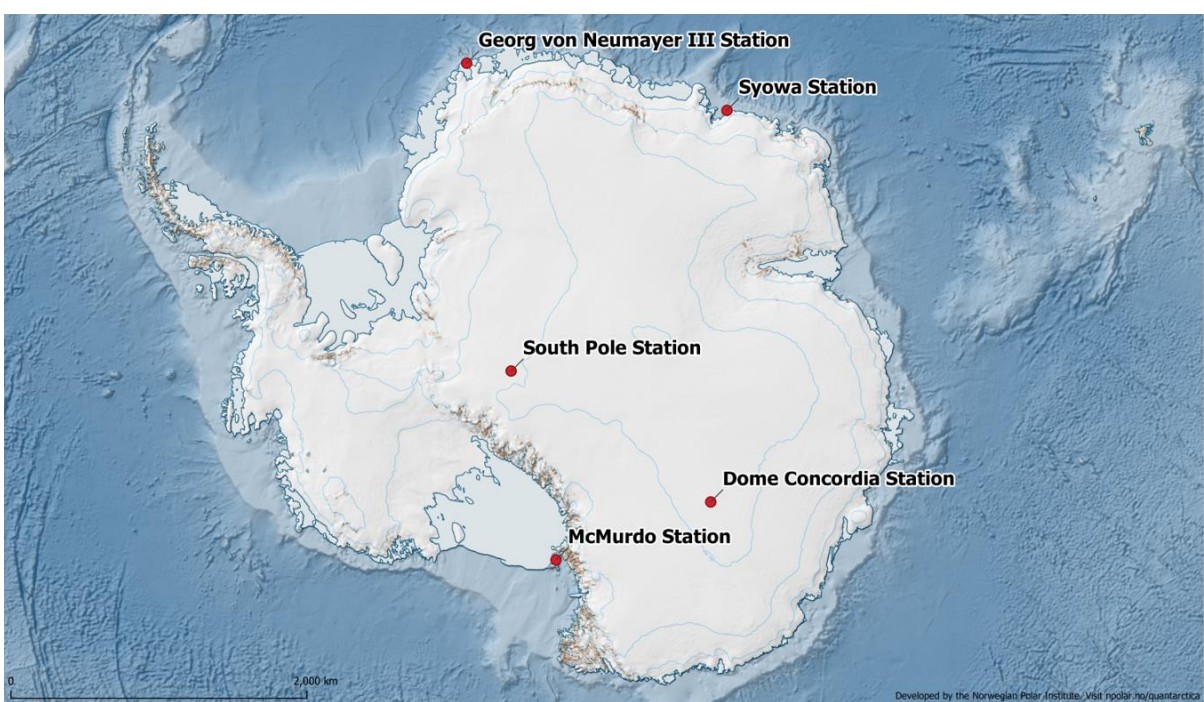

*Figure 1: As seen in Dice et al. (2023): Map locations of all study sites (red dots with station names)*
*across the continent. Map courtesy of Quantarctica (Matsuoka et al., 2018).*

## 2 Data and Methods

### 2.1 Data

Radiosonde data from two continental interior sites (South Pole and Dome C) and three coastal sites (McMurdo, Neumayer, and Syowa) (Figure 1, Table 1) as well as corresponding downwelling longwave radiation data at the time the radiosonde launches occurred are included in this analysis. As described by Dice et al. (2023), the lengths of data sets (13 months at McMurdo to 19 years at Syowa) are used for the data presented here, reflecting the longest term, continuous, and easily accessible data set from each of the five sites listed above. The shorter period of data from McMurdo is used to coincide with availability of radiosonde and radiation data from the year-long Department of Energy Atmospheric Radiation Measurement (ARM) West Antarctic Radiation Experiment (AWARE) (Lubin et al., 2017, 2020; Silber et al., 2018) campaign, which was also previously analyzed by Dice and Cassano (2022) and Dice et al. (2023). The data from Neumayer station is also a relatively shorter data set, as data with high enough resolution was not available until 2018 (Dice et al., 2023). The data sets at South Pole, Dome C, and Syowa have data available spanning more than at least 15 years.

Located 2835 m above sea level, South Pole is a high elevation continental interior site known for its persistent cold conditions and temperature inversions (Zhang et al., 2011; Hudson and Brandt, 2005; etc.). Radiosonde data from 1 January 2005 through 29 September 2021 have been retrieved from the Antarctic Meteorological Research and Data Center (AMRDC). The radiosondes at South Pole are launched once per day at 2100 UTC, and twice per day during the austral summer when conditions allow. The radiative flux data from South Pole are from the Baseline Surface Radiation network (BSRN), and the instrumentation for this data is located 0.8 km away from the radiosonde launch site (Table 1).

Also located at high elevation, at 3233 m above sea level, Dome C is located on a plateau with a nearly flat surface around it, characterized by almost constant near-surface temperature inversions and strong stability (Genthon et al., 2013; Pietroni et al., 2013). Radiosonde data from Dome C between 21 January 2006 and 14 October 2021 are from the Antarctic Meteo-Climatological Observatory, and radiosonde launches are performed once per day throughout the year at 1200 UTC. Radiation data from Dome C was obtained from BSRN, and the site of the radiation instrumentation is located 0.6 km away from the radiosonde launch site (Table 1).

Located on Hut Point Peninsula of Ross Island, McMurdo is a coastal site surrounded by complex topography, where Mt. Erebus rises to 3,794 m. McMurdo is located between McMurdo Sound to the west and north and the Ross Ice Shelf to the south and east. The data used in this study from McMurdo are from the AWARE campaign (Lubin et al., 2017, 2020; Silber et al., 2018), which occurred at McMurdo from 20 November 2015 to 3 January 2017. During AWARE, radiosonde launches occurred twice daily at 1000 UTC and 2200 UTC. The surface radiative flux data from AWARE were recorded approximately 2 km away from the radiosonde launch site (Table 1). The radiosonde site is characterized by coastal influences from McMurdo Sound, with slower wind speeds and warmer temperatures, whereas higher wind speeds and colder temperatures are characteristic of the higher elevation observation site on the Ross Ice Shelf side of the Hut Point peninsula where the surface radiation was measured during AWARE (Dice and Cassano, 2022)

Located near sea-level on the Ekström Ice Shelf, Neumayer is characterized by flat and homogeneous terrain. Neumayer is influenced by cyclone activity in the circumpolar trough, which can act to quickly impact boundary layer stability at this site (Silva et al., 2022). Radiosonde and surface radiative flux data from Neumayer Station are from BSRN, recorded from 1 June 2018 to 31 January 2021, with radiosonde launches occurring once per day at 1200 UTC, and when conditions are favorable

during austral summer, a second launch occurs at 0500 UTC. The site of the instrumentation for the
radiative flux data is located 3.1 km away from the radiosonde launch site (Table 1).
Syowa station is located near sea level on East Ongul Island in the Lutzow-Holm Bay, where the
wind and weather conditions are impacted by cyclone activity and katabatic winds from the continental
interior (Murakoshi, 1958; Yamada and Hirasawa, 2018). Radiosonde data from 1 February 2001 through
23 January 2020 are from the Office of Antarctic Observation Japan Meteorological Agency (pers. comm.
Yutaka Ogawa). The radiosonde launches occur twice pre day at 1130 UTC and 2330 UTC. The surface
radiative flux data is from BSRN, and the instrumentation for this data is located 1.1. km away from the
radiosonde launch site (Table 1).
The radiosonde observations from all five sites will be analyzed from 20 m above ground level
(AGL) to 500 m AGL. The height of 20 m was chosen as the lowest height to analyze, as oftentimes,
warm biases near the surface in radiosonde data are observed below this height, due to radiosondes being
moved from warm buildings to outside without enough time to equilibrate to outside temperatures before
launch (Schwartz and Doswell, 1991; Mahesh et al., 1997). The height of 500 m was chosen to be the top
of the profiles we will analyze here, as the depth of the boundary layer was below 500 m in most cases
(Dice and Cassano, 2022; Dice et al., 2023). The boundary layer stability profiles in this study will be
assessed based on the vertical potential temperature gradient from each radiosonde profile.
Given the two separate locations of the radiosonde launch sites and the surface observation site, it
is important to note that these two locations could have slightly different meteorological conditions. For
this reason, and because several of the sites have different heights at which surface wind speed is
recorded, the surface wind speeds discussed in this study will be near-surface 20 m wind speeds taken
from the radiosonde observations rather than surface wind speeds from the respective surface observation
sites.
*Table 1: Information for each of the five study sites: South Pole, Dome C, McMurdo, Neumayer, and*
*Syowa. From left to right, the columns indicate: study site, latitude, longitude and elevation above sea*
*level (ASL), site location type, distance between the location of the radiosonde launches and the location*
*of the surface observation instrumentation, the type of radiosonde and accuracy of the temperature and*
*wind measurements, respectively, the radiation instrumentation and accuracy, the time period of the*
*radiosonde and radiation data, and the number of radiosonde launches in the dataset.*

| Station | Latitude, Longitude, Elevation | Site Type | Distance between Observations | Radiosonde Type and Accuracy | Radiation Instrument and Accuracy | Time Period of Surface Observations | Number of Profiles |
|---|---|---|---|---|---|---|---|
| South Pole | -89.98°S, 24.80°W; 2,836 m | Interior plateau | 811.8 m | Vaisala RS41-SGP radiosondes; 0.2 K, 0.5 m s$^{-1}$ | Pyrgeometer, Eppley, PIR; 5 W m$^{-2}$ | 01 Jan 2005-29 Sep 2021 | 8,587 |
| Dome Concordia | -75.10°S, 123.33°E; 3,251 m | Interior plateau | 571.8 m | RS-92 radiosondes; 0.2 K, 0.2 m s$^{-1}$ | Pyrgeometer, Kipp & Zonen, CG4; <7.5 W m$^{-2}$ | 21 Jan 2006-14 Oct 2021 | 5,147 |
| McMurdo | -89.98°S, 24.80°W; 2,836 m | Coastal; Ross Island | 1.7 km | RS-92 radiosondes; 0.2 K, 0.2 m s$^{-1}$ | Pyrgeometer, Eppley, PIR; 5 W m$^{-2}$ | 30 Nov 2015-03 Jan 2017 | 8,587 |
| Georg von Neumayer | -70.65°S, -8.17°W; 38 m | Coastal; Ekström Ice Shelf | 3.1 km | Vaisala, RS41-SGP radiosondes; 0.2 K, 0.5 m s$^{-1}$ | Pyrgeometer, Eppley, PIR; 5 W m$^{-2}$ | 01 Jun 2018-31 Jan 2021 | 1,220 |
| Syowa | -69.00°S, 39.58°W; 18.4 m | Coastal; East Ongul Island | 1.1 km | Meisei RS-11G radiosondes; 0.5 K, 2 m s$^{-1}$ | Pyrgeometer, Kipp & Zonen, CG4; <7.5 W m$^{-2}$ | 01 Feb 2007-23 Jan 2020 | 6,390 |

**2.2 Methods**
**2.2.1 Definition Scheme for Boundary Layer Stability Regimes**
Boundary layer stability regimes, accounting for both near-surface stability and stability above
the boundary layer, were defined by Dice et al. (2023) (Table 2) and used to classify the stability in
individual radiosonde profiles. The potential temperature gradient between 20 m and 50 m in each
radiosonde profile were used to define six near-surface stability regimes. These six near-surface stability
regimes range from near neutral conditions (NN; $d\theta/dz < 0.5$ K (100 m)$^{-1}$) to extremely strongly stable
conditions (ESS; $d\theta/dz > 30$ K (100 m)$^{-1}$). Thresholds to distinguish between these six regimes, near
neutral (NN), weak stability (WS), moderate stability (MS), strong stability (SS), very strong stability
(VSS) and extremely strong stability (ESS) were defined by Dice et al. (2023) and Jozef et al. (2023)
(Table 2), and were found to have robust applications in both the Antarctic and Arctic.
Stability regimes aloft, just above the boundary layer, were also defined, as many of the
radiosonde profiles have enhanced stability above layers of weaker, near-surface stability. It is important
to identify the stability structure both within, and just above the boundary layer for understanding of its
evolution in time. For example, enhanced stability above the boundary layer could act to suppress the
growth of the boundary layer with strong radiative forcing or mechanical mixing. Stability aloft was
defined by first finding the top of the boundary layer based on the bulk Richardson number, as described
in Jozef et al. (2022). A ratio between the production or suppression of turbulence by buoyancy and
turbulence generated by wind shear, the bulk Richarson number is used to identify the point in each
radiosonde profile where turbulence is no longer sustained (Stull, 1988). Thus, the height of the boundary
layer is given by the height at which the bulk Richardson number exceeds the critical value (0.5) and
remains above this value for at least 20 consecutive meters in each radiosonde profile. Then, the stability
regime above the boundary layer was found by identifying the maximum potential temperature gradient
between the top of the boundary layer and 500 m (the top of the profile used in this study), using the same
potential temperature gradient thresholds used to define the near-surface stability (Table 2). An aloft
stability regime was only attributed to a radiosonde profile when stability aloft was greater than the near-
surface stability. In cases where the greatest stability in the profile occurs near the surface, no aloft
stability regime is defined.
It was also noted in the near neutral (NN) and weak stability (WS) regimes that there was one
grouping of profiles where the boundary layer depth is greater than 125 m, and one grouping where the
depth was less than 125 m. For these profiles with a boundary layer depth less than 125 m and a NN or
WS stability designation, the regime was instead identified to be very-shallow mixed, or VSM.
The near-surface and aloft stability (if applicable) for each radiosonde profile were combined to
give the final stability regime. Thus, profiles with, for example, near-neutral stability near the surface and
moderate stability above the boundary layer was named as "near-neutral, moderate stability aloft", or
"NN-MSA". Applying this method to the various combinations of near-surface and aloft stability regimes
left seven "stability groupings", where the near-surface stability is the same, but varied stability is present
aloft. For example, the NN "stability grouping" consists of the following NN (near-neutral), NN-WSA
(near-neutral, weak stability aloft), NN-MSA (near-neutral, moderate stability aloft), and NN-SSA (near-
neutral, strong stability aloft). Figures throughout this paper use distinct colors for each of these stability
groupings: NN, brown, VSM, red, WS, green, MS, blue, SS, purple, VSS, pink, and ESS, indigo. The
darkest color in each group is the "basic near-surface stability regime", where no enhanced stability aloft
is present, and the color used to represent the regimes decreases in intensity as stability aloft in each
grouping increases. The basic near-surface stability regimes consist of the following: NN, WS, MS, SS,
VSS, and ESS, as well as VSM-WSA. The VSM-WSA regime is also considered a basic near-surface
stability regime because the VSM portion of this regime is a subset from the NN or WS regime, as it has
the same potential temperature gradient, just a shallower boundary layer (Dice et al., 2023). Additionally,
to help with visualization of the vertical structure of the regimes, an example profile of the potential
temperature gradient and potential temperature anomaly for each of the twenty boundary layer regimes
can be seen in Figure 2.
*Table 2: As seen in Dice et al. (2023): Boundary Layer Regime definition scheme. The left column of the*
*table shows the potential temperature gradient (dθ/dz in K (100 m)$^{-1}$) thresholds used to define each of*
*the six basic near-surface stability regimes from 20 m to 50 m. The middle column shows how the very*
*shallow mixed layer definition was applied to NN and WS cases. The third column shows the maximum*
*potential temperature gradient thresholds ($\mathrm{d}\theta/\mathrm{d}z$ in K (100 m)$^{-1}$) for the aloft stability regimes.*

| Near-Surface Stability | Very Shallow Mixed Layer | Stability Above Boundary Layer ("Aloft") |
|---|---|---|
| **Near-Neutral (NN):** $\mathrm{d}\theta\,\mathrm{d}z^{-1} < 0.5$ K (100 m)$^{-1}$ | If near-surface stability = NN or WS and ABL height <125 m <br> ➤ Near-surface stability =**Very-Shallow Mixed (VSM)** | |
| **Weak Stability (WS):** $\mathrm{d}\theta\,\mathrm{d}z^{-1} >= 0.5$ K (100 m)$^{-1}$ and $< 1.75$ K (100 m)$^{-1}$ | | **Weak Stability Aloft (-WSA):** $\mathrm{d}\theta\,\mathrm{d}z^{-1} >= 0.5$ K (100 m)$^{-1}$ and $< 1.75$ K (100 m)$^{-1}$ |
| **Moderate Stability (MS):** $\mathrm{d}\theta\,\mathrm{d}z^{-1} >= 1.75$ K (100 m)$^{-1}$ and $< 5$ K (100 m)$^{-1}$ | | **Moderate Stability Aloft (-MSA):** $\mathrm{d}\theta\,\mathrm{d}z^{-1} >= 1.75$ K (100 m)$^{-1}$ and $< 5$ K (100 m)$^{-1}$ |
| **Strong Stability (SS):** $\mathrm{d}\theta\,\mathrm{d}z^{-1} >= 5$ K (100 m)$^{-1}$ and $< 15$ K (100 m)$^{-1}$ | | **Strong Stability Aloft (-SSA):** $\mathrm{d}\theta\,\mathrm{d}z^{-1} >= 5$ K (100 m)$^{-1}$ |
| **Very Strong Stability (VSS):** $\mathrm{d}\theta\,\mathrm{d}z^{-1} >= 15$ K (100 m)$^{-1}$ and $< 30$ K (100 m)$^{-1}$ | | **Very Strong Stability Aloft (-VSSA):** $\mathrm{d}\theta\,\mathrm{d}z^{-1} >= 15$ K (100 m)$^{-1}$ |
| **Extremely Strong Stability (ESS):** $\mathrm{d}\theta\,\mathrm{d}z^{-1} >= 30$ K (100 m)$^{-1}$ | | **Extremely Strong Stability Aloft (-ESSA):** $\mathrm{d}\theta\,\mathrm{d}z^{-1} >= 30$ K (100 m)$^{-1}$ |

*Table 3: As seen in Dice et al. (2023): Boundary Layer Regime acronyms and color codes. On the left is*
*the color and acronym used to represent each of the 20 stability regimes in figures and tables throughout*
*this paper, and the full regime name is spelled out on the right. The basic near-surface stability regimes*
*are denoted in bold font.*

| Regime Color and Acronym | Regime Full Name |
|---|---|
| **NN** | **Near Neutral** |
| NN-WSA | Near Neutral- Weak Stability Aloft |
| NN-MSA | Near Neutral- Moderate Stability Aloft |
| NN-SSA | Near Neutral- Strong Stability Aloft |
| **VSM-WSA** | **Very Shallow Mixed- Weak Stability Aloft** |
| VSM-MSA | Very Shallow Mixed- Moderate Stability Aloft |
| VSM-SSA | Very Shallow Mixed- Strong Stability Aloft |
| **WS** | **Weak Stability** |
| WS-MSA | Weak Stability- Moderate Stability Aloft |
| WS-SSA | Weak Stability- Strong Stability Aloft |
| **MS** | **Moderate Stability** |
| MS-SSA | Moderate Stability- Strong Stability Aloft |
| **MS-VSSA** | **Moderate Stability- Very Strong Stability Aloft** |
| MS-ESSA | Moderate Stability- Extremely Strong Stability Aloft |
| **SS** | **Strong Stability** |
| SS-VSSA | Strong Stability- Very Strong Stability Aloft |
| SS-ESSA | Strong Stability- Extremely Strong Stability Aloft |
| **VSS** | **Very Strong Stability** |
| VSS-ESSA | Very Strong Stability- Extremely Strong Stability Aloft |
| **ESS** | **Extremely Strong Stability** |

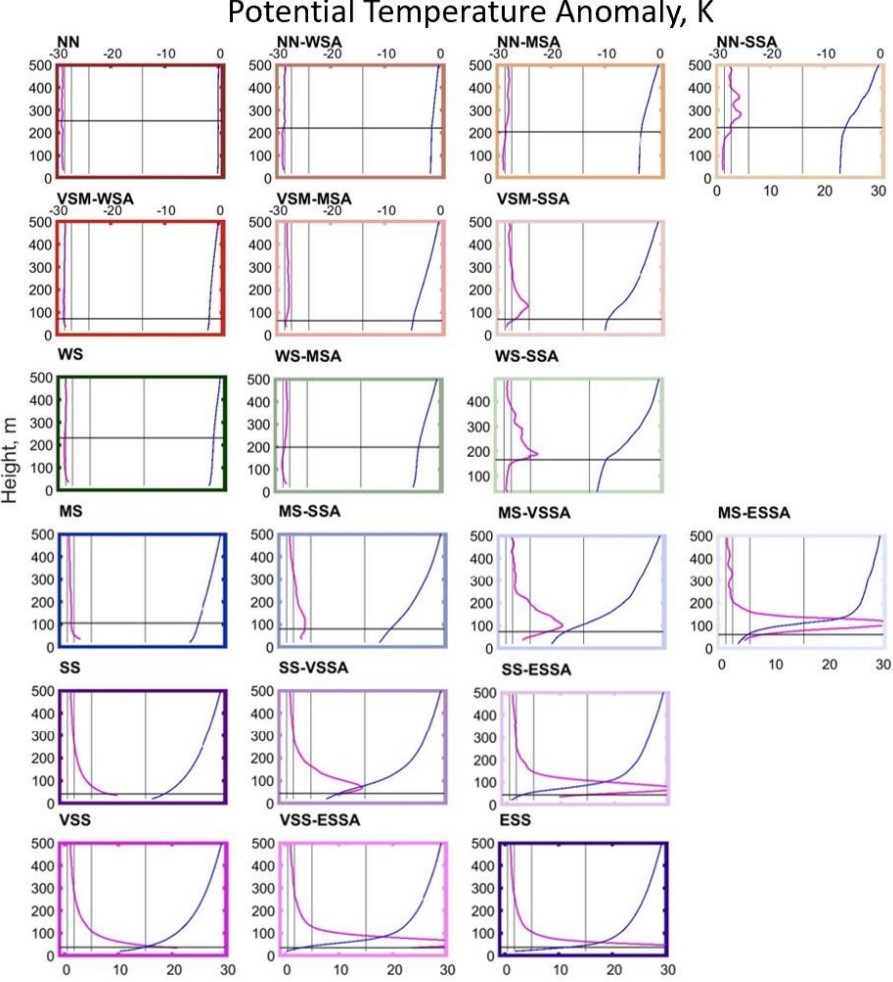

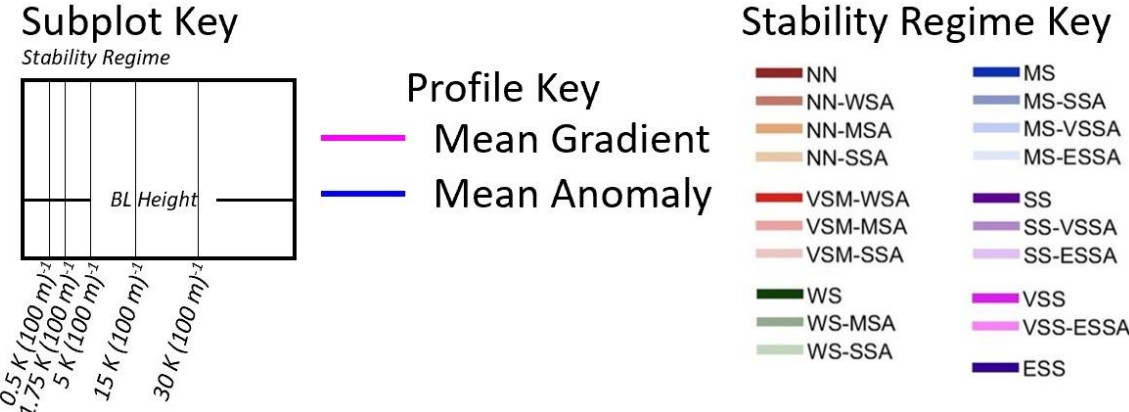

*Figure 2: Examples of the vertical profile structure of the regimes listed in Table 3. The potential temperature gradient is shown in pink (top axis), the potential temperature anomaly (with respect to the 20 m potential temperature form the radiosonde) is shown in blue (bottom axis). The stability regime acronym is given above the top left corner of each subplot and is also indicated by the colored outline around each plot, according to the key in the bottom right of the figure.*

**3 Results**

Once each radiosonde profile has been assigned a boundary layer stability regime, the list of dates
and times when each regime occurred is used to calculate statistics of the boundary layer forcing
mechanisms for each regime. To assess the possible atmospheric forcing that drives the variability in
stability regimes we compare downwelling longwave radiation and 20 m wind speed across the different
stability regimes (Table 2). The 20 m radiosonde wind speed is used rather than the surface wind speed to
remove any potential discrepancy in wind speeds due to the difference in location of surface observations
and radiosonde launch sites, as described in Section 2.1. These two forcing variables serve as proxies for
varying surface energy fluxes and mechanical mixing which may lead to variations in near-surface
stability (Rodrigo and Anderson, 2013). As observed in Dice and Cassano (2022) and other studies,
surface heating or reduced cooling (increased downward radiative fluxes) and increased mechanical
mixing (greater near surface wind speed and shear) lead to weaker stability, while surface cooling and
decreased mechanical mixing allow stable conditions and temperature inversions to form at the surface
(e.g., King and Turner, 1997; Andreas et al., 2000; Hudson and Brandt, 2005). While downwelling
longwave radiation is largely independent of stability, wind speeds can change in response to changes in
stability. For example, with very strong near-surface stability, winds can become decoupled from the
frictional, slowing effects of the surface and increase. In addition to these two variables, additional
forcing mechanisms, such as the passing of synoptic cyclones or other weather systems, or low-level
advection, all of which could result in changes in near surface stability, are possible, although not
investigated at length in this analysis.

Box plots of downwelling longwave radiation and 20 m radiosonde wind speed are shown for
each stability regime with increasing stability, from NN to ESS, from left to right on an annual (left panel)
and seasonal (right four panels) basis at each site (Figures 3 through 12). The seasons are defined in this
study as follows: summer (DJ), fall (FMA), winter (MJJA), and spring (SON), as used in previous studies
of the Antarctic (Cassano et al., 2016, Seefeldt and Cassano, 2012). Each box plot shows the mean (black
asterisk), median (black horizontal line), $25^{th}$ and $75^{th}$ percentiles (edges of box), and $10^{th}$ and $90^{th}$
percentiles (whiskers) for each regime, although the analysis below will primarily focus on the mean
values. The number of observations in each regime annually and seasonally are given by the numbers at
the top of each plot, and the horizontal black line across each of the annual and seasonal panels is the
mean for that period of time. Regimes with fewer than 10 observations will not be discussed at length, as
these small sample sizes may not be representative. The number of observations, mean downwelling
longwave radiation, and mean 20 m wind speed in each regime are listed in Tables S1 to S5 for each site.

**3.1 South Pole**

South Pole, a high-plateau continental interior site, is generally characterized by strong and
persistent radiative cooling allowing for the formation of strong stability (Stone and Kahl, 1991; Lazzara
et al., 2012). Dice et al. (2023) noted that boundary layer stability at South Pole was largely dominated by
the SS, VSS, or ESS regimes, occurring near the surface 51% of the year. However, they found when
considering the maximum stability in the profile, SS, VSS, or ESS conditions occur 85.2% of the year,
either near the surface or aloft but below 500 m. Here, the radiative forcing and mechanical mixing for
each stability regime will be analyzed on an annual and seasonal basis.

The downwelling longwave radiation for each regime annually and seasonally is shown in Figure
3 and Table S1. Considering changes in downwelling longwave radiation as stability increases, the first
result to note is that annually, the downwelling longwave radiation (Figure 3) decreases by nearly half
from weak to strong stability across the basic near-surface stability regimes from NN (174 W m$^{-2}$) to ESS
(91 W m$^{-2}$). Similarly, in the fall and spring, downwelling longwave radiation consistently decreases from
the MS (138 W m$^{-2}$ in the fall 129 W m$^{-2}$ in the spring) to ESS (95 W m$^{-2}$ in the fall and 93 W m$^{-2}$ in the
spring) basic near-surface stability regimes, which are the most common regimes in these seasons. In the
winter downwelling longwave radiation is higher in the SS (115 W m$^{-2}$) regime, compared to the much
lower values in the VSS (90 W m$^{-2}$) and ESS (88 W m$^{-2}$) regimes, indicating a clear difference in forcing
for relatively weaker versus relatively stronger stability regimes. A similar observation is noted in the
summer, where the downwelling longwave radiation is similar in the NN (173 W m$^{-2}$) and VSM-WSA
(163 W m$^{-2}$) regimes and is then about 21% lower, and similar, across the WS, MS, and SS regimes,
ranging from 128 to 137 W m$^{-2}$. Annually and across all seasons, the downwelling longwave radiation in
the SS, VSS, and ESS regimes is almost always lower than the seasonal mean, and the downwelling
longwave radiation in the NN regime, and usually in the VSM regime, is above the seasonal mean.

It is also important to note the influence of downwelling shortwave radiation in the summer and
transition seasons, as enhanced downwelling shortwave radiation can also reduce near-surface stability.
On an annual basis at South Pole downwelling shortwave radiation across the NN, VSM-WSA, WS, and
MS basic near-surface stability regimes is highest (362 W m$^{-2}$ to 394 W m$^{-2}$) and above the annual mean,
then dramatically decreases in the SS regime (248 W m$^{-2}$) and is lowest and below the annual mean in the
VSS regime (76 W m$^{-2}$) and the ESS regime, which occurs almost exclusively when downwelling
shortwave radiation is zero (Figure S1). These results show that the strongest stability regimes can only
form when there is very little downwelling shortwave radiation. With downwelling shortwave radiation
much higher than 300 W m$^{-2}$ throughout the summer season, it is thus not surprising that the strongest
stability regimes (VSS and ESS) occur rarely or not at all. In the fall and spring, during the transiting into
or out of the polar night, a wide range of downwelling shortwave radiation is possible and a strong
decrease in downwelling shortwave radiation is noted going from the MS regime (233 W m$^{-2}$ in the fall
and 341 W m$^{-2}$ in the spring) to the ESS regime (24 W m$^{-2}$ in the fall and 62 W m$^{-2}$ in the spring) which
further supports the observation that these strongest stability regimes are limited to periods with little or
no sunlight.

When stability aloft increases within a given stability grouping, downwelling longwave radiation
usually decreases. In the fall, winter, and spring, and on an annual basis, this decrease is largest in the MS
stability grouping, by as much as 26 W m$^{-2}$ to 37 W m$^{-2}$, followed by the decrease in the SS stability
grouping, by as much as 16 W m$^{-2}$ to 26 W m$^{-2}$. In the summer, downwelling longwave radiation
decreases within the NN (15 W m$^{-2}$), VSM (16 W m$^{-2}$), and MS (6 W m$^{-2}$) stability groupings, although
not as strongly as what was seen within stability groupings in the other seasons, while downwelling
longwave radiation is more similar as stability aloft increases in the WS stability grouping.

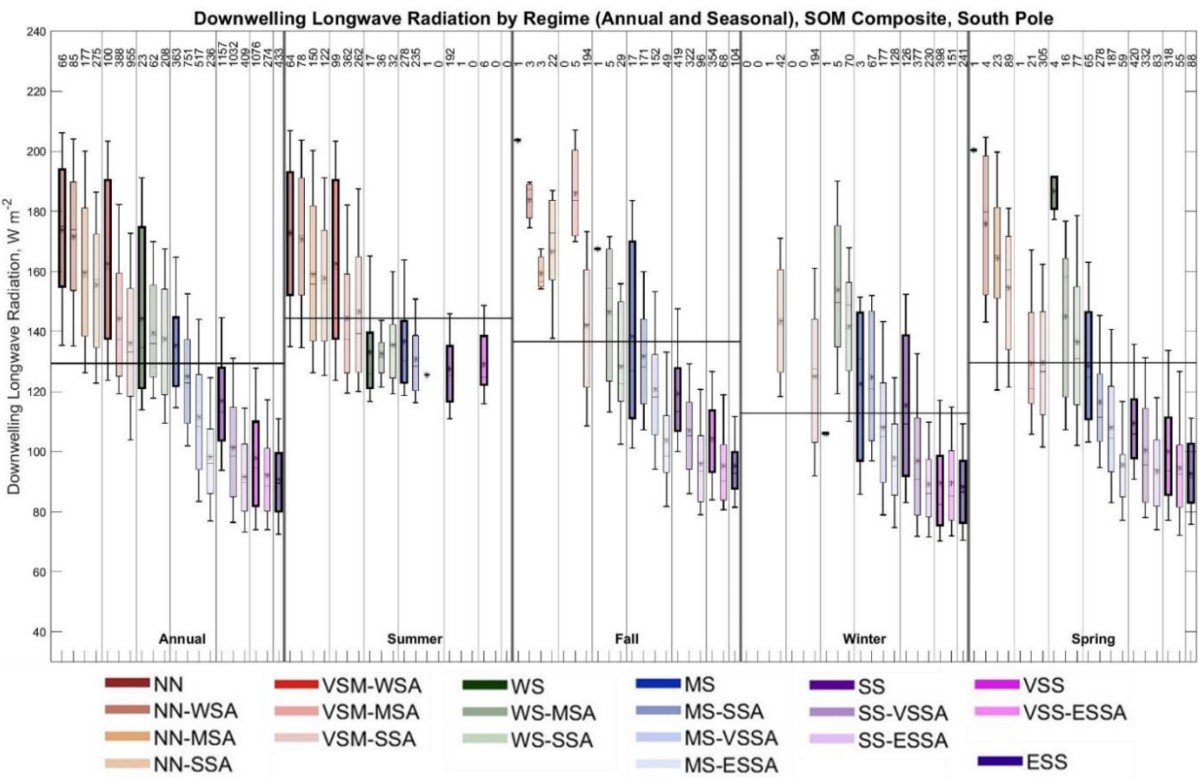

*Figure 3: Box plot showing the distribution of downwelling longwave radiation observed for each stability regime at*
*South Pole annually (left panel) and seasonally (right four panels – summer, fall, winter, and spring). Box plots*
*show median downwelling longwave radiation (horizontal line), mean downwelling longwave radiation (black star),*
*25th and 75th percentiles (edges of boxes), and 10th and 90th percentiles (whiskers). The thin vertical black lines in*
*the figure separate the stability groupings in each panel (annual or seasonal). The thin horizontal black lines across*
*each panel (annual or seasonal), indicate the mean value for that entire time period. The numbers at the top indicate*
*the number of radiosonde profiles in each regime.*

While the trend in downwelling longwave radiation both annually and seasonally generally shows
a clear decrease from weak to strong stability both at the surface and aloft (Figure 3 and Table S1), the 20
m wind speed (Figure 4; Table S1) observations for the various regimes shows less of a clear difference in
wind speed with varying stability. However, it is noted that for most near-surface stability groupings the
20 m wind speed tends to increase with increasing stability aloft, suggesting that increased mechanical
mixing by stronger winds is required for maintaining reduced near-surface stability as stability aloft
increases, consistent with Dice and Cassano (2022).
When looking at just the basic near-surface stability regimes on an annual basis mean wind
speeds are highest in the WS regime (6.2 m s$^{-1}$), and then lower and similar across the SS, VSS, and ESS
regimes, ranging from 5.1 m s$^{-1}$ to 5.4 m s$^{-1}$, and lowest in the NN, VSM-WSA, and MS regimes, ranging
from 4.3 m s$^{-1}$ to 4.5 m s$^{-1}$. A similar pattern is observed in the summer, where wind speeds are the
strongest in the WS regime (5.7 m s$^{-1}$), weaker in the NN (4.5 m s$^{-1}$) and the VSM-WSA (4.3 m s$^{-1}$)
regimes and weakest in the MS (3.9 m s$^{-1}$) and SS regimes (3.7 m s$^{-1}$). Annually and in the summer, the
stronger wind speeds in the WS regime in comparison to the VSM-WSA regime is a key difference that
distinguishes these regimes which are similar in boundary layer strength (Table 2) but have different
boundary layer depths. This will be discussed further in the discussion section below. In the fall and
spring, wind speed is slightly higher in the MS regime (6.0 m s$^{-1}$ in the fall and 5.8 m s$^{-2}$), and weaker and
similar across the SS, VSS and ESS regimes (between 4.7 m s$^{-1}$ and 5.2 m s$^{-1}$ in the fall and between 5.0
m s$^{-1}$ and 5.2 m s$^{-2}$ in the spring). In the winter, wind speeds decrease from SS (7.1 m s$^{-1}$) to ESS (5.3 m s$^{-1}$), which is a more consistent decrease with increasing stability, and more like the expected result that
weaker winds are associated with stronger stability (e.g., Cassano et al., 2016).

Winds generally increase with increasing stability aloft in each stability grouping annually and, in

the fall and spring (Figure 4; Table S1). Annually, the wind speed increases the most in the MS stability
grouping from 4.4 m s$^{-1}$ to 8.5 m s$^{-1}$, but also shows clear increases across the NN, VSM, WS, SS, and
VSS stability groups. Wind speed increases 1.3 m s$^{-1}$ to 2.8 m s$^{-1}$ with increasing stability in the
frequently observed MS, SS, and VSS stability groups in the fall and spring. In contrast, in the winter, as
stability aloft increases within stability groupings, wind speed increases only slightly with increasing
stability aloft in the SS (0.6 m s$^{-1}$) and VSS (0.5 m s$^{-1}$) stability groupings. In comparison, wind speeds
across the MS-SSA, MS-VSSA, and MS-ESSA regimes do not follow a very clear trend, ranging from
8.5 m s$^{-1}$ to 8.9 m s$^{-1}$. Similarly in the summer, wind speed increases 0.6 m s$^{-1}$ to 0.9 m s$^{-1}$ in the NN,
VSM, and WS stability groupings as stability aloft increases. It is also interesting to note that the mean
wind speed for the basic near-surface stability regimes, annually and seasonally, is generally lower than
the annual or seasonal mean, while the wind speeds in regimes with enhanced stability aloft is often
higher than the annual or seasonal mean. As noted above, this suggests that stronger mechanical mixing
may be needed to reduce near-surface stability in the presence of enhanced stability aloft, which was also
noted by Dice and Cassano (2022) at McMurdo.

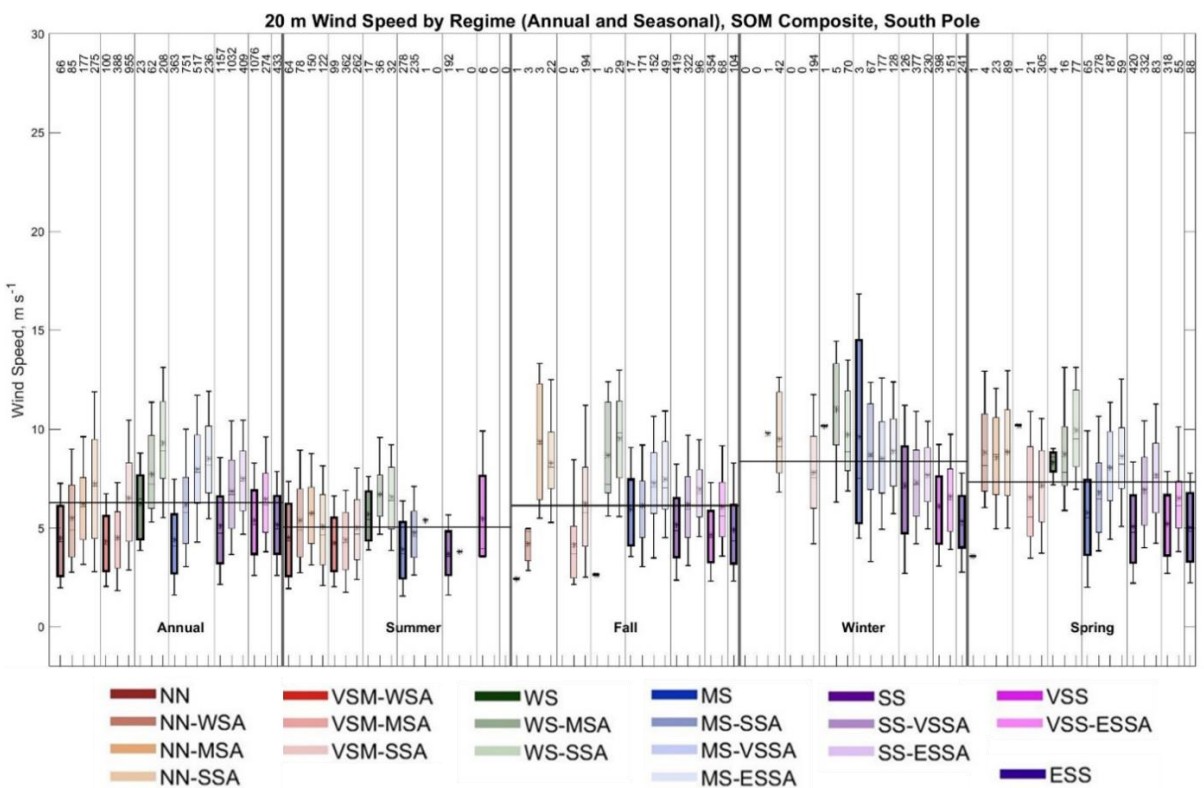

*Figure 4: Box plot showing the distribution of 20 m wind speed observed for each stability regime at South Pole*
*annually (left panel) and seasonally (right four panels – summer, fall, winter, and spring). Box plots show median*
*20 m wind speed (horizontal line), mean 20 m wind speed (center black star), 25th and 75th percentiles (edges of*
*boxes), and 10th and 90th percentiles (whiskers). The thin vertical black lines in the figure separate the stability*
*groupings in each panel (annual or seasonal). The thin horizontal black lines across each panel (annual or*
*seasonal), indicate the mean value for that entire time period. The numbers at the top indicate the number of*
*radiosonde profiles in to each regime.*

### 3.2 Dome C

At Dome C, strong radiative cooling throughout the year and associated strong surface temperature inversions are well documented (Phillpot and Zillman, 1970; Hudson and Brandt, 2005; Genthon et al., 2013; Ganeshan et al., 2022; Dice et al., 2023; etc.). Like South Pole, Dice et al. (2023) observed consistently strong stability (SS, VSS, and ESS regimes) throughout the year at Dome C, occurring near the surface 73.6% of the year and 82.4% of the time either at the surface or just above the boundary layer.

Downwelling longwave radiation observed for each stability regime, both annually and seasonally, is shown in Figure 5 and Table S2. Considering first changes in downwelling longwave radiation across just the basic near-surface stability regimes, a clear decrease of downwelling longwave radiation occurs as basic near-surface stability increases annually and, in the fall, winter, and spring. Across the basic near-surface stability regimes downwelling longwave radiation decreases by nearly half from the VSM-WSA (123 W m$^{-2}$) to the ESS (79 W m$^{-2}$) regimes on an annual basis. While only MS and stronger regimes are observed regularly during fall, winter, and spring, there is also a clear decrease in downwelling longwave radiation from MS (116 W m$^{-2}$ in the fall, 132 W m$^{-2}$ in the winter, and 104 W m$^{-2}$ in the spring) to ESS (83 W m$^{-2}$ in the fall, 76 W m$^{-2}$ in the winter, and 81 W m$^{-2}$ in the spring) regimes during these seasons. During these times of the year, downwelling longwave radiation is generally less than the annual or seasonal means for the SS and stronger stability regimes, and usually greater than the annual or seasonal means in MS or weaker stability regimes. In summer, there is little change in downwelling longwave radiation across the most frequently observed basic-near surface stability regimes (VSM-WSA to VSS) ranging from 120 W m$^{-2}$ to 127 W m$^{-2}$.

Similar to South Pole, downwelling shortwave radiation is much higher in the basic near-surface stability regimes of NN, VSM-WSA, WS, and MS (557 W m$^{-2}$ to 616 W m$^{-2}$) on an annual basis in comparison to in the SS (199 W m$^{-2}$), VSS (127 W m$^{-2}$), and ESS (63 W m$^{-2}$) regimes, further indicating that these regimes mostly form when there is little or no solar radiation (Figure S2). This is also observed in the transition seasons, with downwelling shortwave radiation decreasing sharply from the MS regime (449 W m$^{-2}$ in the fall and 532 W m$^{-2}$ in the spring) to the ESS regime (105 W m$^{-2}$ in the fall and 194 W m$^{-2}$ in the spring), which in combination with the decrease in downwelling longwave radiation, contributes to the range of regimes observed in these seasons. Surprisingly, only a slight decrease in downwelling shortwave radiation occurs across the basic near-surface stability regimes in the summer, from the VSM-WSA regime (616 W m$^{-2}$) to the VSS regime (588 W m$^{-2}$). This suggests that changes in shortwave radiation are likely not important in distinguishing these different stability regimes.

Changes in downwelling longwave radiation within regime groups, as aloft stability increases, is not always as clear as was seen for the near-surface stability regimes (Figure 5; Table S2). On an annual basis there is little change in downwelling longwave radiation within the NN, VSM, or WS stability groups but there is a consistent decrease in downwelling longwave radiation as aloft stability increases in the MS, SS, and VSS stability groups. In fall and spring, downwelling longwave radiation also consistently decreases in the SS and VSS stability groups, and slightly decreases from MS to MS-VSSA in the fall. In the winter, downwelling longwave radiation decreases from MS to MS-VSSA. In some cases, there is little change within regime groups (e.g., SS and VSS in in winter, and MS in the fall and spring, excluding MS-ESSA) while in other cases there is only a noticeable decrease in downwelling longwave radiation for the strongest aloft stability within a regime group (e.g., SS in fall and SS and VSS in spring). In the summer there is little change in downwelling longwave radiation as stability aloft increases within the various stability regime groups.

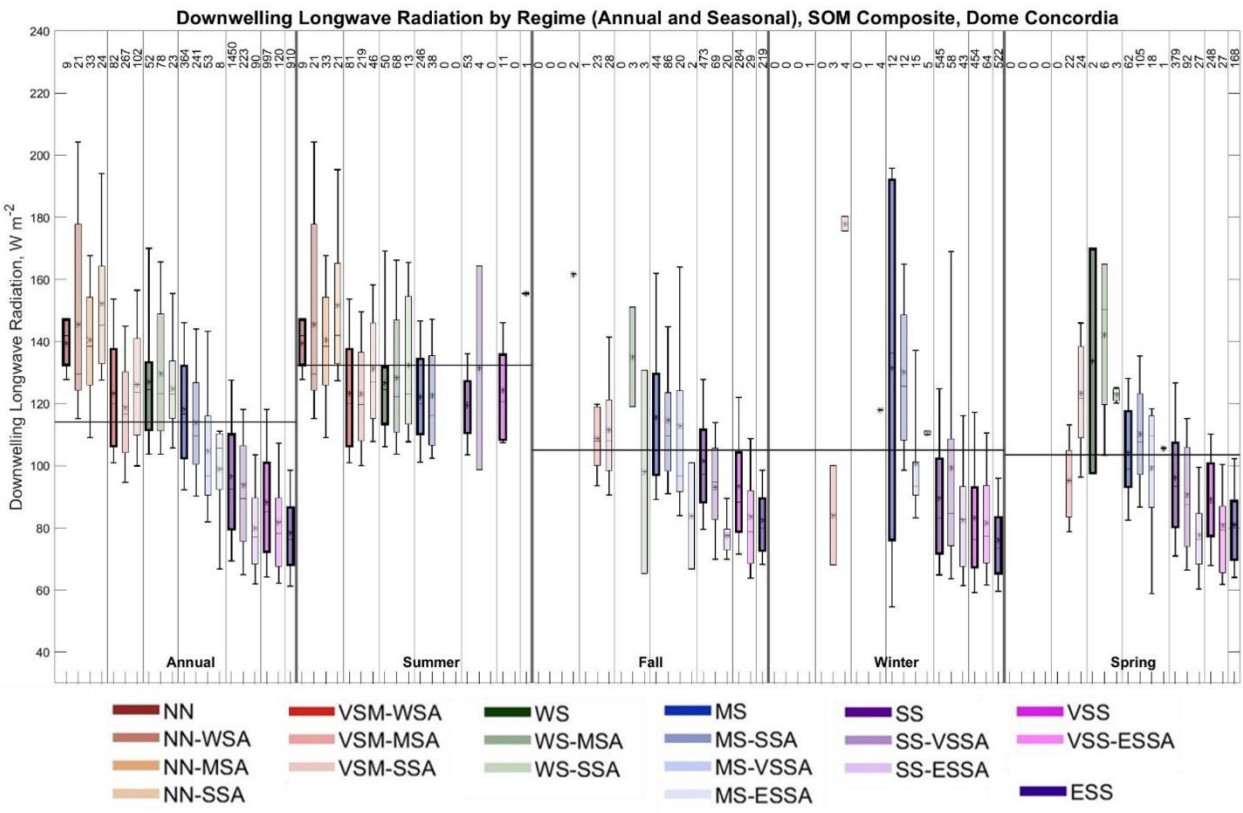

*Figure 5: Box plot showing the distribution of downwelling longwave radiation observed for each stability regime at Dome C annually (left panel) and seasonally (right four panels – summer, fall, winter, and spring). Box plots show median downwelling longwave radiation (horizontal line), mean downwelling longwave radiation (black star), 25th and 75th percentiles (edges of boxes), and 10th and 90th percentiles (whiskers). The thin vertical black lines in the figure separate the stability groupings in each panel (annual or seasonal). The thin horizontal black lines across each panel (annual or seasonal), indicate the mean value for that entire time period. The numbers at the top indicate the number of radiosonde profiles in each regime.*

The distribution of 20 m wind speed for each stability regime, on an annual and seasonal basis, is shown in Figure 6 and Table S2. Interestingly, wind speed generally increases with increasing stability annually and in the fall, winter, and spring, which is unexpected. Another robust feature seen in Figure 6 is that 20 m wind speed generally increases within regime groups as aloft stability increases, such that mean wind speed for the regimes with enhanced stability aloft is often above the annual or seasonal mean, while mean wind speeds for the basic near-surface stability regimes are below or close to the annual or seasonal mean.

Considering first the basic near-surface stability regimes, a surprising result is seen for the annual data. The 20 m wind speed increases by almost 80% from the weakest stability, VSM-WSA (3.3 m s$^{-1}$) to the strongest stability, ESS (7.7 m s$^{-1}$). As discussed in the introduction, stronger winds are typically associated with weaker near-surface stability (e.g., Pietroni et al., 2013; Cassano et al., 2016), thus, this is a surprising result, which will be discussed further in Section 4. In the winter, for the basic near-surface stability regimes with the most observations, the wind speed is highest in the MS regime (9.0 m s$^{-1}$), decreases to SS (5.1 m s$^{-1}$), and then increases to ESS (8.0 m s$^{-1}$). In the fall and spring, the MS (5.3 m s$^{-1}$ in the fall and 5.8 m s$^{-1}$ in the spring) and SS (4.8 m s$^{-1}$ in the fall and 5.5 m s$^{-1}$ in the spring) regimes have similar wind speeds that are below the seasonal mean, while the wind speed is higher in and increases

from VSS (5.9 m s$^{-1}$ in the fall and 6.5 m s$^{-1}$ in the spring) to ESS (7.3 m s$^{-1}$ in both seasons). Unlike what
was seen for the annual data, differences in the 20 m wind speed across the basic near-surface stability
regimes in the summer do not show a consistent pattern as stability varies. The 20 m wind speed is
weakest for the VSM-WSA regime (3.2 m s$^{-1}$), almost 40% stronger and similar for the WS (4.3 m s$^{-1}$),
VSS (4.6 m s$^{-1}$), and SS (4.8 m s$^{-1}$) regimes and strongest for the MS (5.3 m s$^{-1}$) regime. The weaker
winds in the VSM-WSA regime in comparison to those in the WS regime will be discussed in detail in
Section 4.

When looking at wind speed variability within stability groups as stability aloft increases there is a
relatively consistent pattern of stronger winds being associated with increasing stability aloft. This is very
clearly seen in the annual data in Figure 6 and Table S2. Here, the wind speed change is largest in the SS
regime group, increasing from 5.1 m s$^{-1}$ to 10.1 m s$^{-1}$, and in MS regime group, increasing from 5.5 m s$^{-1}$
to 9.8 m s$^{-1}$. Smaller increases in wind speed, of 1.8 m s$^{-1}$ to 2.8 m s$^{-1}$, are seen across the NN, VSM, WS,
and VSS regime groups annually.  Clear increases in wind speed with increasing stability aloft are seen
for the MS, SS and VSS regime groups in fall, winter, and spring. The largest increase in wind speed
occurs in the SS regime in the winter (increase of 6.1 m s$^{-1}$) and fall (3.8 m s$^{-1}$) and in the MS regime in
the spring (increase of 4.4 m s$^{-1}$). In the summer, speeds weakly increase in the NN, WS and MS regimes
(0.5 m s$^{-1}$ to 1.4 m s$^{-1}$) and show little change for the other regime groups. In most cases the mean wind
speed in each regime is less than the annual or seasonal mean for the basic near-surface stability regimes
and increases to greater than the annual or seasonal mean for many of the enhanced stability aloft
regimes. This suggests that to maintain a given near-surface stability stronger winds, and mechanical
mixing, is required as stability aloft increases. This behavior is consistent with findings of Cassano et al.
(2016), Dice and Cassano (2022) and others that found that stronger winds typically reduce near surface
stability.

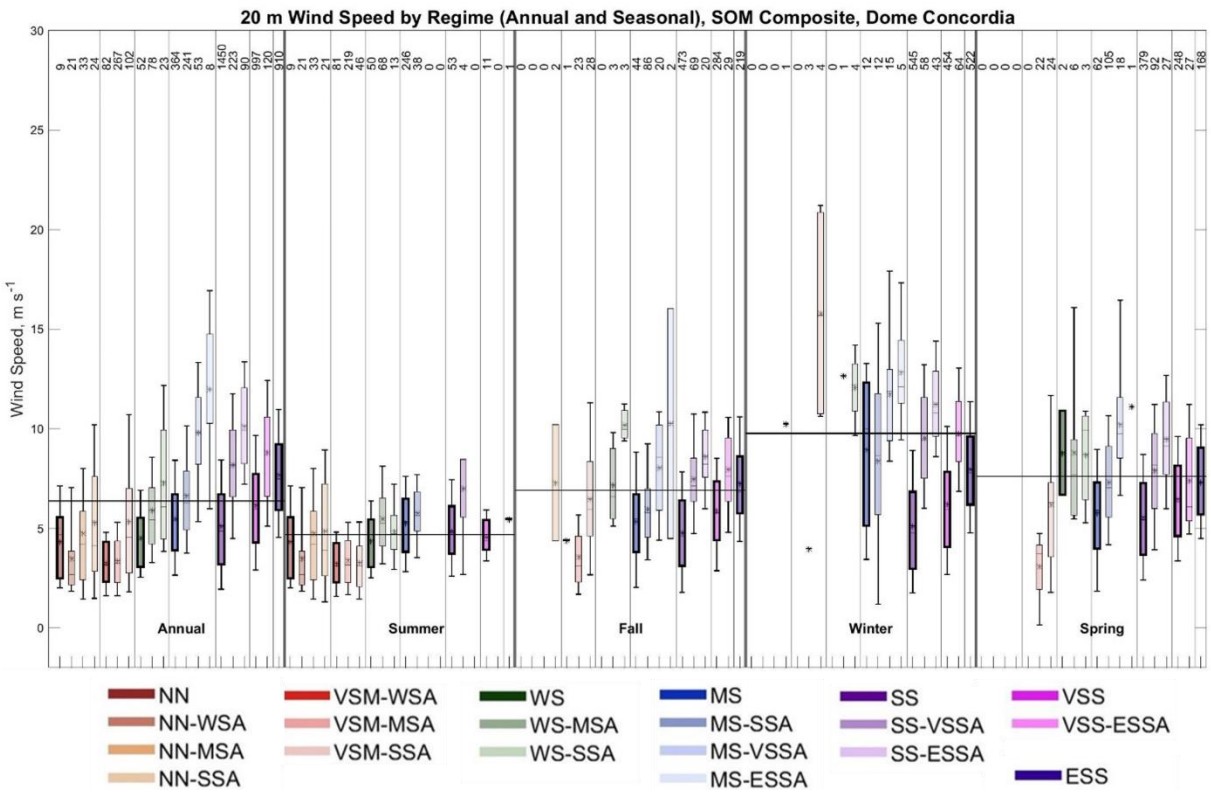

*Figure 6: Box plot showing the distribution of 20 m wind speed observed for each stability regime at Dome C*
*annually (left panel) and seasonally (right four panels – summer, fall, winter, and spring). Box plots show median*
*20 m wind speed (horizontal line), mean 20 m wind speed (center black star), 25th and 75th percentiles (edges of*
*boxes), and 10th and 90th percentiles (whiskers). The thin vertical black lines in the figure separate the stability*
*groupings in each panel (annual or seasonal). The thin horizontal black lines across each panel (annual or*
*seasonal), indicate the mean value for that entire time period. The numbers at the top indicate the number of*
*radiosonde profiles in each regime.*
**3.3 McMurdo**

The results at the two continental interior sites above are reflective of the nearly constant,
strongly stable conditions seen in the boundary layer throughout much of the year there, that form in
response to the extremely low values of downwelling longwave radiation (Phillpot and Zillman, 1970;
Zhang, et al., 2011; Dice et al., 2023). Now, the three coastal sites will be analyzed: McMurdo,
Neumayer, and Syowa. In comparison to the continental interior sites, a wider range of boundary layer
stability regimes are present at these sites (Dice et al., 2023), and are expected to have more complex
forcing mechanisms, such as temperature advection (Dice and Cassano, 2022), katabatic winds
(Murakoshi, 1958; Hudson and Brandt, 2005; Lazzara et al., 2012), and cyclonic activity (Silva et al.,
2022). Specifically at McMurdo, Dice et al. (2023) found that the summer was largely dominated by the
NN, VSM, and WS regimes (92.1%), while near surface stability in the winter was more varied but found
that MS or SS conditions occur near the surface or aloft 84.6% of the winter season.

The downwelling longwave radiation at McMurdo as a function of stability regime is shown
annually and seasonally in Figure 7 and Table S3. Most notably, the downwelling longwave radiation
shows a clear decrease from weak to strong stability across the basic near-surface stability regimes
annually and in the transition seasons. On an annual basis downwelling longwave radiation decreases by
over 70 W m$^{-2}$ from NN (232 W m$^{-2}$) to SS (161 W m$^{-2}$). In the transition seasons, the decrease from

weakest to strongest stability is between 16 W m$^{-2}$ and 36 W m$^{-2}$, from VSM-WSA (191 W m$^{-2}$ in the fall and 200 W m$^{-2}$ in the spring) to SS (175 W m$^{-2}$ in the fall and 164 W m$^{-2}$ in the spring). There is not a consistent decrease in downwelling longwave radiation with increasing basic near-surface stability in the summer or winter for the most frequently observed regimes. In the summer, downwelling longwave radiation is highest in the NN basic near-surface stability regime (244 W m$^{-2}$), slightly less in the MS regime (235 W m$^{-2}$) and lowest in the VSM-WSA regime (227 W m$^{-2}$). In the winter, downwelling longwave radiation is about the same in the MS regime (148 W m$^{-2}$) and the SS regime (149 W m$^{-2}$). Generally, across all seasons and annually, regimes with stability MS and stronger have downwelling longwave radiation below the seasonal mean (Figure 7). These results are consistent with those found by Dice and Cassano (2022) at McMurdo Station, where decreasing downwelling longwave radiation with increasing stability was observed annually and seasonally, with the highest values observed in summer, and lowest in winter.

In addition, downwelling shortwave radiation at McMurdo (Figure S3) is higher in the NN and VSM-WSA (204 W m$^{-2}$ to 207 W m$^{-2}$) basic near-surface stability regimes in comparison to the WS, MS, SS, and VSS stability regimes (7 W m$^{-2}$ to 123 W m$^{-2}$) annually. This pattern is also observed for the regimes present in the fall and spring (VSM-WSA, and MS and SS). These results suggest that reductions in both downwelling longwave and shortwave radiation result in increased near surface stability. A less clear pattern emerges in the summer, where downwelling shortwave radiation is lower in the NN regime (264 W m$^{-2}$) and higher in the VSM-WSA and MS regimes (both 350 W m$^{-2}$), and thus other factors are likely more important in distinguishing regimes in this season.

When considering stability aloft, downwelling longwave radiation usually decreases with increasing stability aloft (Figure 7; Table S3). Annually, downwelling longwave radiation decreases within the NN (45 W m$^{-2}$ difference) and VSM (24 W m$^{-2}$) stability groups, and within the VSM stability group in the spring (28 W m$^{-2}$) and from NN-MSA to NN-SSA in the winter (27 W m$^{-2}$). In most of the other stability groupings in the other seasons, downwelling longwave radiation decreases only slightly as aloft stability increases (e.g.in the NN regime group in summer and fall) or does not show a uniform change as aloft stability increases (e.g., WS regime group in winter and spring).

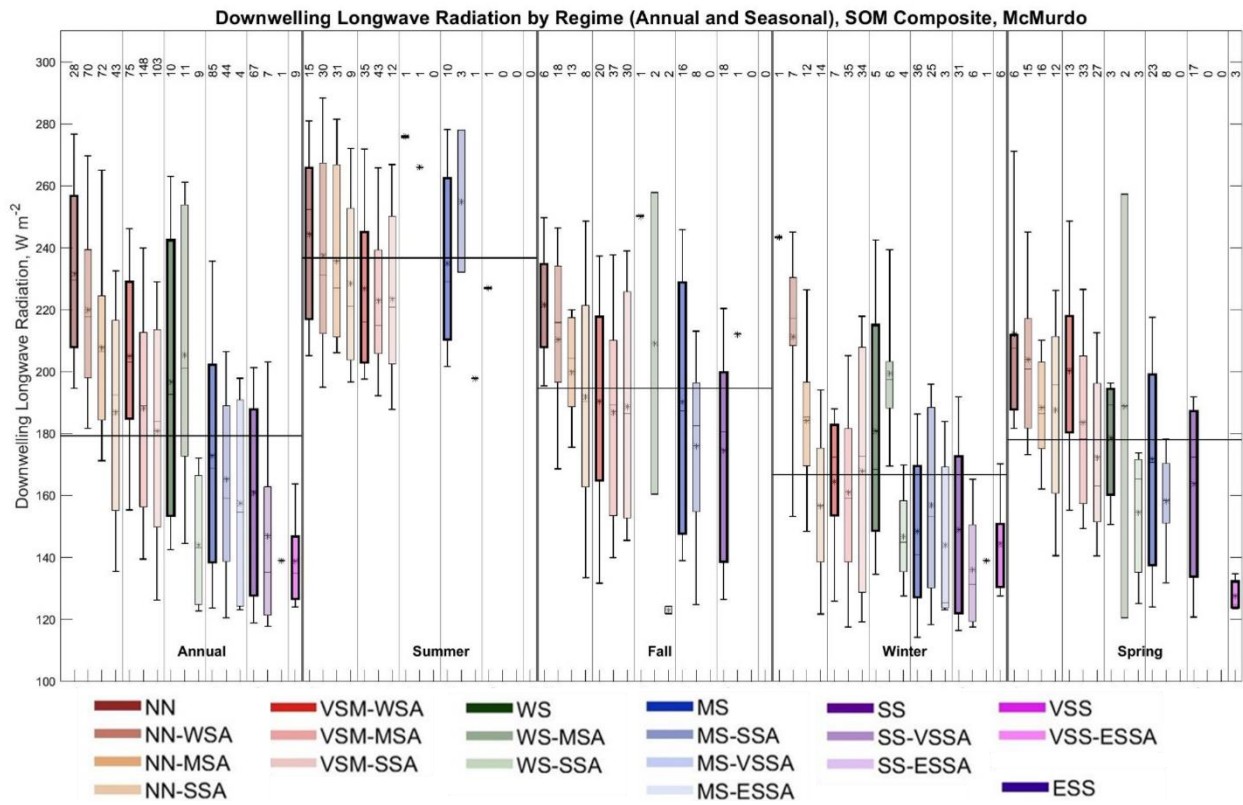

Figure 7: Box plot showing the distribution of downwelling longwave radiation observed for each stability regime at
McMurdo annually (left panel) and seasonally (right four panels – summer, fall, winter, and spring). Box plots show
median downwelling longwave radiation (horizontal line), mean downwelling longwave radiation (black star), 25th
and 75th percentiles (edges of boxes), and 10th and 90th percentiles (whiskers). The thin vertical black lines in the
figure separate the stability groupings in each panel (annual or seasonal). The thin horizontal black lines across
each panel (annual or seasonal), indicate the mean value for that entire time period. The numbers at the top indicate
the number of radiosonde profiles in each regime.

Considering now the 20 m wind speed at McMurdo annually and seasonally (Figure 8; Table S3),
there is not a clear pattern across the basic near-surface stability regimes, but there is a tendency for wind
speed to increase with increasing stability aloft in many of the stability groups.

Annually, wind speed is greatest in the WS (5.3 m s$^{-1}$) and NN (5.2 m s$^{-1}$) basic near-surface
stability regimes. Wind speeds are more than 2 m s$^{-1}$ lower and similar across the VSM-WSA, MS, and
SS regimes (2.5 m s$^{-1}$ to 3.0 m s$^{-1}$). Similarly, in the summer the wind speed is highest in the NN regime
(5.4 m s$^{-1}$) and more than 3 m s$^{-1}$ less in the VSM-WSA and MS regimes. The weaker winds in the VSM-
WSA regime, compared to either the NN or WS regimes will be discussed further in the next section.
Winds are similar between the frequently observed MS (1.9 m s$^{-1}$) and SS (2.0 m s$^{-1}$) regimes in winter. In
the fall similar winds occur between VSM-WSA (3.0 m s$^{-1}$) and MS (2.8 m s$^{-1}$), then increase from MS to
SS (5.3 m s$^{-1}$). In the spring, similar wind speeds also occur between VSM-WSA (3.1 m s$^{-1}$) and MS (3.3
m s$^{-1}$), but then decrease from MS to SS (2.4 m s$^{-1}$).

Wind speed increases with increasing stability aloft, for each stability grouping, on an annual
basis and usually in the seasons as well (Figure 8; Table S3). The largest change in wind speed with
increased stability aloft occurs in the fall. At this time of year wind speeds within the NN and VSM
regimes increase by over half between the basic near-surface stability regime and the strongest aloft

stability regime (3.9 m s$^{-1}$ to 6.5 m s$^{-1}$ from NN-WSA to NN-MSA and 3.0 m s$^{-1}$ to 7.1 m s$^{-1}$ in the VSM stability grouping). Wind speed also generally increases with increasing stability aloft within each stability group for the other seasons, but usually by less than 2 m s$^{-1}$, and often closer to 1 m s$^{-1}$. This tendency for wind speed to increase with increasing stability aloft was also noted at both the continental interior sites above and may reflect the need for stronger winds to weaken the near surface stability when stronger stability aloft is present. The exceptions to this are decreases, rather than increases, in wind speed of 1 to 2 m s$^{-1}$ with increasing aloft stability in the NN stability group in the winter (NN-MSA to NN-SSA; a decrease of 1.1 m s$^{-1}$) and spring (NN-WSA to NN-SSA; a decrease of 0.6 m s$^{-1}$).

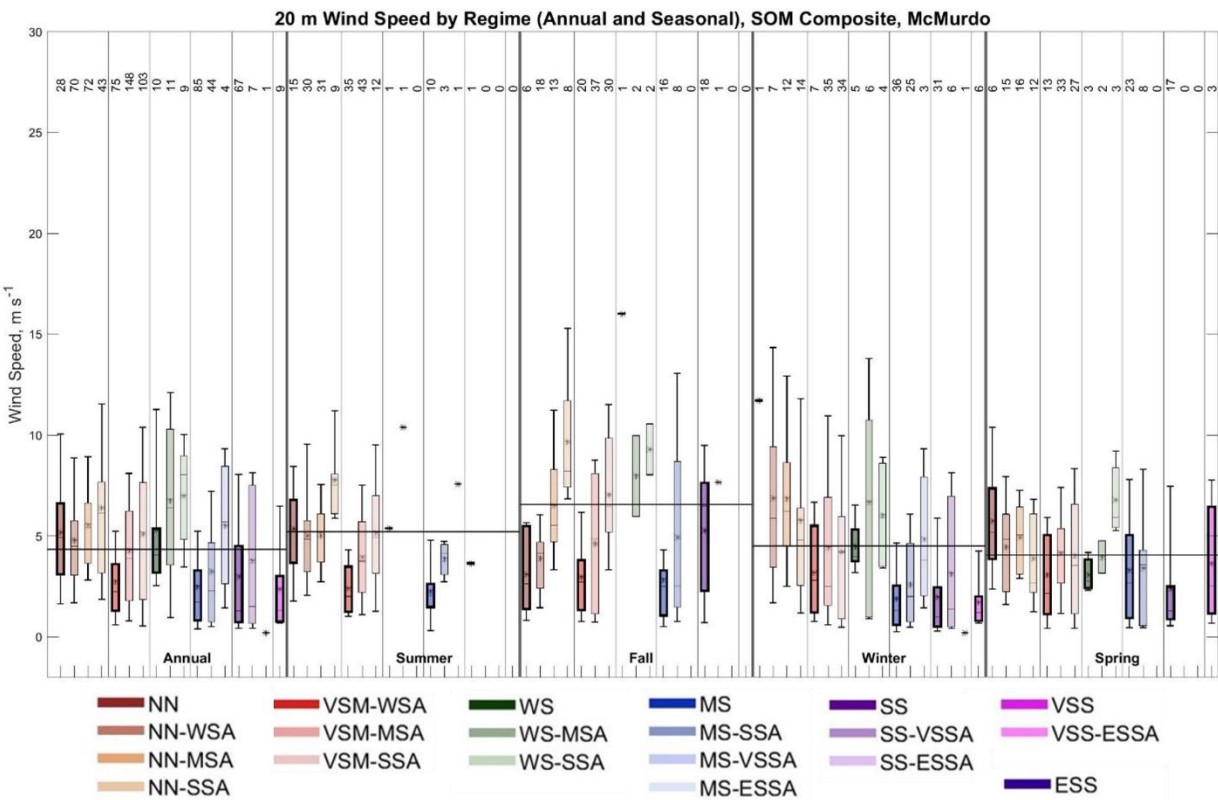

*Figure 8: Box plot showing the distribution of 20 m wind speed observed for each stability regime at McMurdo annually (left panel) and seasonally (right four panels – summer, fall, winter, and spring). Box plots show median 20 m wind speed (horizontal line), mean 20 m wind speed (center black star), 25th and 75th percentiles (edges of boxes), and 10th and 90th percentiles (whiskers). The thin vertical black lines in the figure separate the stability groupings in each panel (annual or seasonal). The thin horizontal black lines across each panel (annual or seasonal), indicate the mean value for that entire time period. The numbers at the top indicate the number of radiosonde profiles in each regime.*

**3.4 Neumayer**

Like McMurdo, a wide range of boundary layer regimes, compared to the near-constant strong stability at South Pole and Dome C, is present at Neumayer (Dice et al., 2023). Neumayer is another coastal site located on an ice shelf and is often influenced by the passing of cyclones, which impacts stability in the boundary layer and results in quickly changing meteorological conditions (Silva et al., 2022). Dice et al. (2022) found boundary layer stability regime distribution similar to that of McMurdo, with the summer largely characterized by NN, VSM, and WS regimes (80.1%). In the winter, moderate or strong stability, either near the surface or aloft, above a layer of weaker stability is often present (85.2%).

Figure 9 and Table S4 show the range of downwelling longwave radiation across stability regimes annually and seasonally at Neumayer. The first thing to note is that downwelling longwave radiation generally decreases with increasing stability across the basic near-surface stability regimes annually and seasonally. The largest decrease is seen in the spring from the NN to SS regime, with a difference of 164 W m$^{-2}$. Decreases on the order of 40 W m$^{-2}$ are observed in the summer (41 W m$^{-2}$), fall (43 W m$^{-2}$) and winter (44 W m$^{-2}$). While there is a general trend for downwelling longwave radiation to decrease from weakest to strongest stability regimes, in the summer and winter the weakest stability regimes (NN (summer only), VSM-WSA and WS) have similar values of downwelling longwave radiation that is noticeably larger than for the stronger stability regimes. This suggests that there may be fundamental differences in radiative forcing between weaker and stronger stability regimes in these seasons.

Additionally, the downwelling shortwave radiation (Figure S4) decreases consistently from the NN (302 W m$^{-2}$) to the VSS (15 W m$^{-2}$) basic near-surface stability regimes on an annual basis. In the summer, downwelling shortwave radiation in the VSM-WSA regime (302 W m$^{-2}$) is much lower than that in the NN (385 W m$^{-2}$) and WS regimes (407 W m$^{-2}$), consistent with less radiative forcing and a shallower boundary layer in the VSM-WSA regime. A decrease in downwelling shortwave radiation from the WS regime (407 W m$^{-2}$) to MS regime (306 W m$^{-2}$) in combination with the decrease in downwelling longwave radiation also appears to contribute to the differences in stability in these regimes in summer. Similarly, downwelling shortwave radiation decreases from the NN (258 W m$^{-2}$) to SS (128 W m$^{-2}$) regimes in the spring and from VSM-WSA (219 W m$^{-2}$) to SS (83 W m$^{-2}$) regimes in the fall indicating that changes in downwelling shortwave radiation likely contribute to the changing stability.

A comparison of downwelling longwave radiation across stability regimes can also be made as stability aloft increases within a given stability regime grouping. The most noteworthy observation is the very strong decrease within stability groupings as stability aloft increases in the spring where downwelling longwave radiation decreases by as much as 42 W m$^{-2}$ in the NN stability grouping and 23 W m$^{-2}$ in the MS stability grouping. A weaker decrease is observed in the fall for the MS (16 W m$^{-2}$) and VSM (7 W m$^{-2}$) stability groups. In the summer, downwelling longwave radiation decreases with increasing stability in the VSM and MS stability groupings, but not in the NN or WS groupings, where downwelling longwave radiation is more varied. A similar observation is noted for the winter, where downwelling longwave radiation slightly decreases in the NN stability grouping (excluding the basic near-surface stability regime of NN), is nearly the same within the MS stability grouping, and increases or is variable in all the other stability groupings.

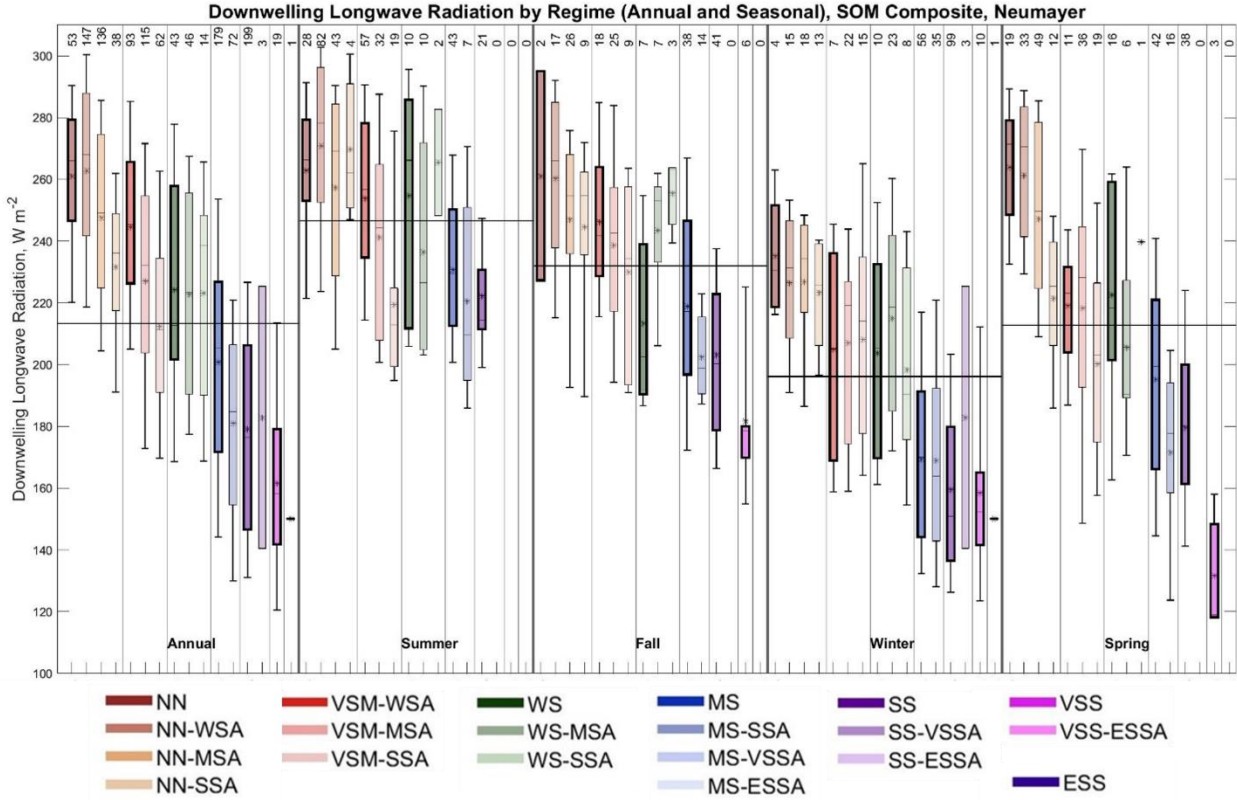

*Figure 9: Box plot showing the distribution of downwelling longwave radiation observed for each stability regime at*
*Neumayer annually (left panel) and seasonally (right four panels – summer, fall, winter, and spring). Box plots*
*show median downwelling longwave radiation (horizontal line), mean downwelling longwave radiation (black star),*
*25th and 75th percentiles (edges of boxes), and 10th and 90th percentiles (whiskers). The thin vertical black lines in*
*the figure separate the stability groupings in each panel (annual or seasonal). The thin horizontal black lines across*
*each panel (annual or seasonal), indicate the mean value for that entire time period. The numbers at the top indicate*
*the number of radiosonde profiles in each regime.*
The 20 m wind speed for each regime annually and seasonally is shown in Figure 10 and Table
S4. Annually wind speeds are highest in the NN (8.9 m s$^{-1}$) and WS (7.9 m s$^{-1}$) basic near-surface stability
regimes and lowest in VSM-WSA (4.3 m s$^{-1}$) regime. Wind speeds are similar in MS and SS (5.0 m s$^{-1}$ to
5.2 m s$^{-1}$) and slightly higher in the VSS (6.0 m s$^{-1}$) regime. The wind speed in the MS, SS and VSS
regimes are higher than those in the VSM-WSA regime but lower than those in the NN and WS regimes.
Similarly, in the summer and spring, wind speeds are highest in NN (7.4 m s$^{-1}$ in the summer and 11.3 m
s$^{-1}$ in the spring ) and WS (8.7 m s$^{-1}$ in the summer and 8.5 m s$^{-1}$ in the spring), and lower and similar
across VSM-WSA, MS, and SS (4.2 m s$^{-1}$ to 5.3 m s$^{-1}$ in the summer and 5.2 m s$^{-1}$ to 5.8 m s$^{-1}$ in the
spring) regimes. In the winter, wind speeds decrease slightly from WS (7.3 m s$^{-1}$) to SS (5.2 m s$^{-1}$) and are
then slightly higher in VSS (6.7 m s$^{-1}$). In the fall, wind speeds are weakest in VSM-WSA (2.8 m s$^{-1}$), and
higher and decrease slightly from MS to SS (4.9 m s$^{-1}$ to 3.7 m s$^{-1}$). It is also interesting to note here that
in most cases in all seasons the VSM-WSA, MS, SS and VSS basic near-surface stability regimes have
wind speeds lower than the seasonal mean, while the NN and WS regimes have mean winds speeds close
to or above the seasonal mean. This observation is consistent with Silva et al. (2022), who observed
weaker wind speeds with stronger stability at Neumayer. Winds in the VSM-WSA regime in comparison
to those in the NN and WS regimes are 49% weaker on an annual basis, and 41% to 47% weaker in the
summer and spring.

When considering stability aloft, another interesting result from Figure 10 and Table S4 is that
wind speed generally increases with increasing stability aloft in the stability groupings annually and
seasonally, although this is usually most evident in the NN, VSM and WS regime groups. As discussed
for other sites above, this may indicate that stronger mechanical mixing is necessary to reduce near-
surface stability. The increase in wind speed with increased stability aloft is largest in the NN regime in
the winter (5.2 m s$^{-1}$) and summer (2.5 m s$^{-1}$) and the VSM regime in the fall (3.6 m s$^{-1}$) and spring (3.9 m
s$^{-1}$. Additionally, regimes with enhanced stability aloft tend to have wind speeds above the seasonal mean,
especially in the NN and WS regime groupings, in comparison to the basic near-surface stability regimes.

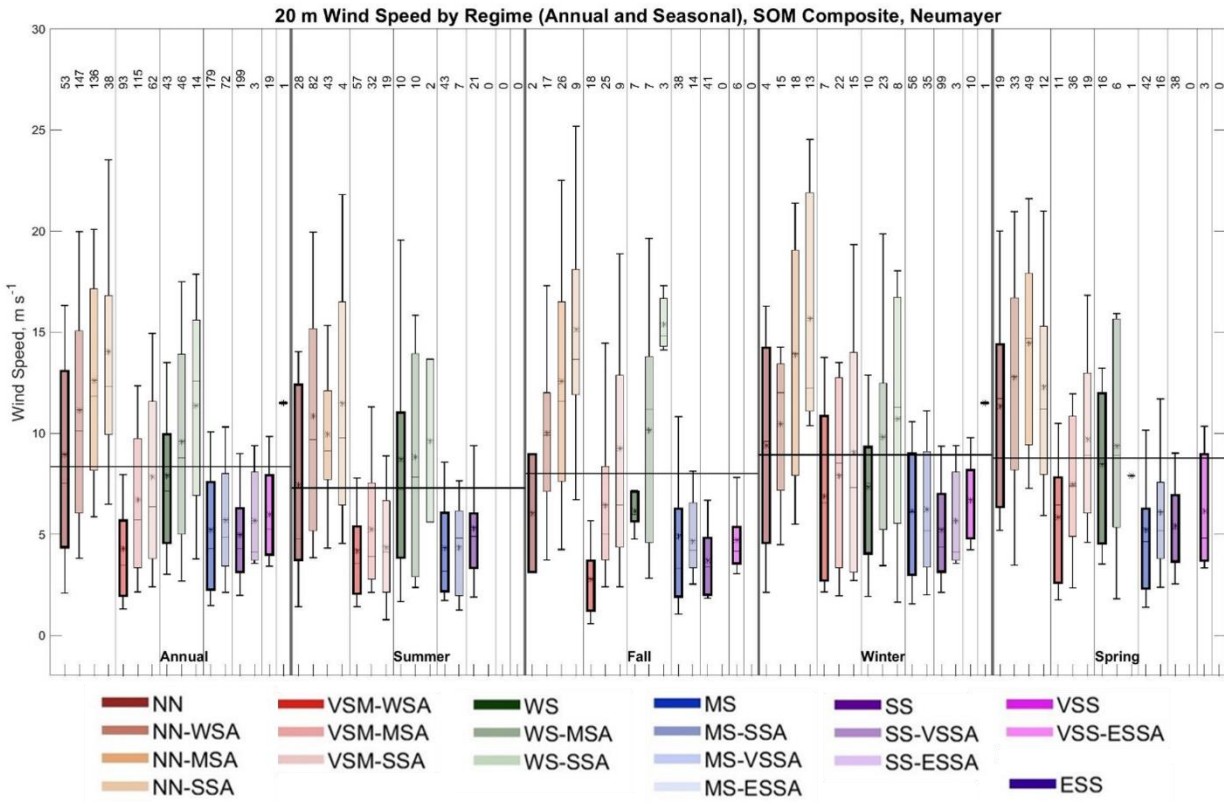

*Figure 10: Box plot showing the distribution of 20 m wind speed observed for each stability regime at Neumayer*
*annually (left panel) and seasonally (right four panels – summer, fall, winter, and spring). Box plots show median*
*20 m wind speed (horizontal line), mean 20 m wind speed (center black star), 25th and 75th percentiles (edges of*
*boxes), and 10th and 90th percentiles (whiskers). The thin vertical black lines in the figure separate the stability*
*groupings in each panel (annual or seasonal). The thin horizontal black lines across each panel (annual or*
*seasonal), indicate the mean value for that entire time period. The numbers at the top indicate the number of*
*radiosonde profiles in each regime.*
**3.5 Syowa**

At Syowa, katabatic winds from the continental interior as well as passing cyclones both impact
boundary layer conditions at this site (Murakoshi, 1958; Yamada and Hirasawa, 2018), resulting in
potentially quickly changing stability. A variety of stability regimes are observed at this site (Dice et al.,
2023), and like the other coastal sites, the summer is largely made up of the NN, VSM, and WS regimes
(82.9%) near the surface. In the winter stronger stability either near the surface or aloft is generally
present (71.1%).

Figure 11 and Table S5 shows the downwelling longwave radiation at Syowa for each regime
annually and seasonally. The first thing to note about the downwelling longwave radiation at Syowa is
that the NN, VSM-WSA, and WS basic near-surface stability regimes have similar and larger
downwelling longwave radiation than the MS and stronger stability regimes annually and for each season.
On an annual basis, mean downwelling longwave radiation varies from 226 W m$^{-2}$ to 236 W m$^{-2}$ across
the NN, VSM-WSA, and WS regimes and then steadily decreases from WS (234 W m$^{-2}$) to VSS (154 W
m$^{-2}$). In the winter and spring, this pattern is the strongest, with downwelling longwave radiation ranging
from 208 W m$^{-2}$ to 236 W m$^{-2}$ across the NN, VSM-WSA, and WS regimes. It is then about 44 W m$^{-2}$
lower in MS (178 W m$^{-2}$ to 184 W m$^{-2}$), SS (168 W m$^{-2}$ to 169 W m$^{-2}$), and VSS (150 W m$^{-2}$). This pattern
is weaker but still present in the summer and fall, with downwelling longwave radiation ranging from 239
W m$^{-2}$ to 253 W m$^{-2}$ across NN, VSM-WSA, and WS in the summer. It is then approximately 26 W m$^{-2}$ to
34 W m$^{-2}$ lower in MS (214 W m$^{-2}$ to 227 W m$^{-2}$) and SS (199 W m$^{-2}$ to 223 W m$^{-2}$). Annually and
seasonally, downwelling longwave radiation in the NN, VSM-WSA, and WS basic near-surface stability
regimes is usually above the seasonal mean while the downwelling longwave radiation in the MS, SS, and
VSS regimes is usually below the seasonal mean. This suggests distinct radiative forcing for the most
stable basic near-surface stability regimes (MS and stronger) compared to the three weakest regimes (NN,
VSM-WSA, and WS) annually and seasonally at Syowa.

On an annual basis, downwelling shortwave radiation at Syowa consistently decreases from the
NN basic near-surface stability regime (203 W m$^{-2}$) to the VSS (53 W m$^{-2}$) basic near-surface stability
regime, and this pattern occurs in the transition seasons as well (Figure S5). In concert with the distinction
in downwelling longwave radiation between the NN, VSM-WSA, WS regimes versus the lower
downwelling longwave radiation in the WS and stronger regimes, this decrease in downwelling shortwave
radiation is likely a contributing factor in distinguishing regimes in the transition seasons. In the summer,
downwelling shortwave radiation is similar in the NN (299 W m$^{-2}$) and VSM-WSA (303 W m$^{-2}$) regimes,
but then sharply decreases to the WS regime (249 W m$^{-2}$). A slight increase in downwelling shortwave
radiation from the WS (249 W m$^{-2}$) to the SS (267 W m$^{-2}$) regimes in the summer is likely counteracted
by the decrease in downwelling longwave radiation across these regimes.

When considering stability aloft in each stability grouping, generally downwelling longwave
radiation decreases as stability aloft increases for most regimes and seasons (Figure 11, Table S5). The
strongest decrease in downwelling longwave radiation occurs in the winter in the WS regime, a decrease
of 36 W m$^{-2}$. In the transition seasons, there is also a strong decrease in downwelling longwave radiation
especially from VSM-WSA (239 W m$^{-2}$ in the fall and 212 W m$^{-2}$ in the spring) to VSM-SSA (213 W m$^{-2}$
in the fall and 187 W m$^{-2}$ in the spring).

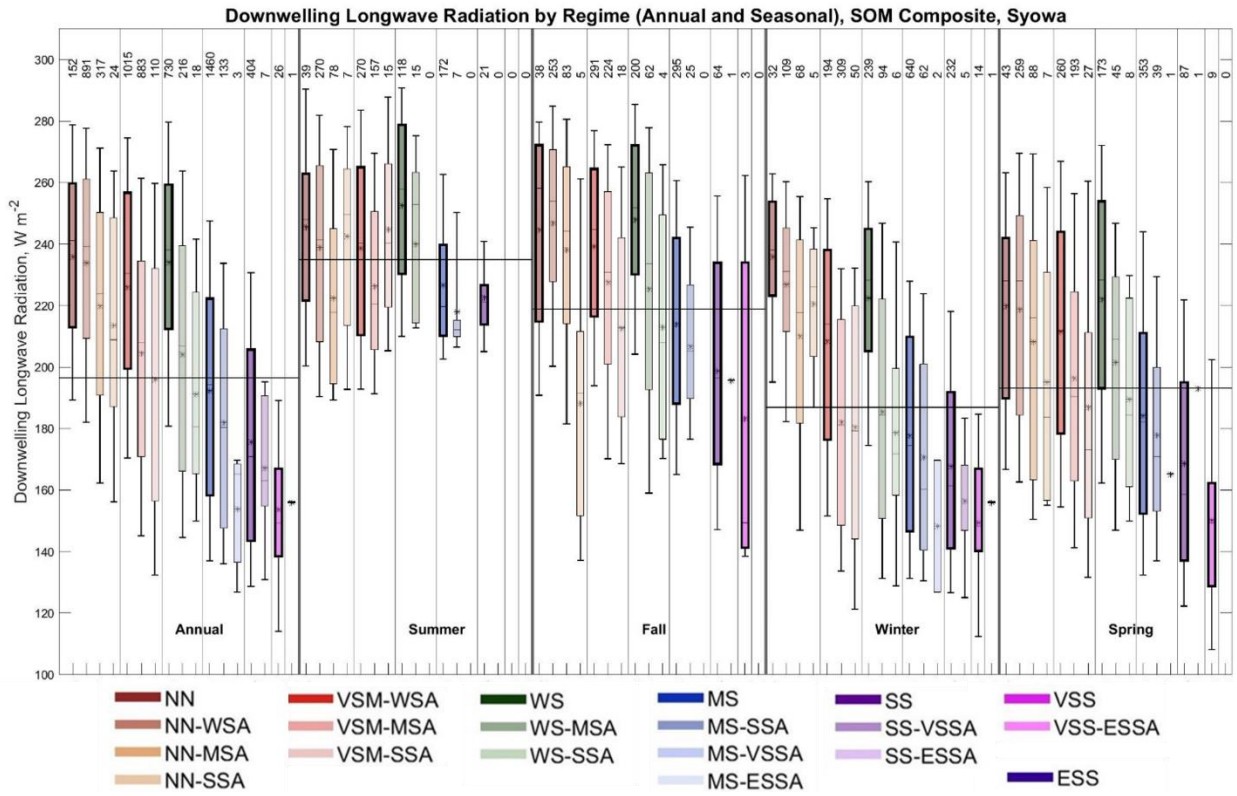

*Figure 11: Box plot showing the distribution of downwelling longwave radiation observed for each stability regime*
*at Syowa annually (left panel) and seasonally (right four panels – summer, fall, winter, and spring). Box plots show*
*median downwelling longwave radiation (horizontal line), 25th and 75th percentiles (edges of boxes), mean*
*downwelling longwave radiation (center black star), 10th and 90th percentiles (outer black stars), and minimum*
*and maximum (whiskers). The thin vertical black lines in the figure separate the stability groupings in each panel*
*(annual or seasonal). The thin horizontal black lines across each panel (annual or seasonal), indicate the mean*
*value for that entire time period. The numbers at the top indicate the number of radiosonde profiles in each regime.*

At Syowa, the 20 m wind speed is shown for each regime on an annual and seasonal basis in
Figure 12 and Table S5. The clearest result from Figure 12 regarding the basic near-surface stability
regimes is the relatively strong wind speeds in the WS regime in comparison to the other regimes,
especially the NN and VSM-WSA regimes, annually and seasonally, except in winter when NN and WS
have similar strong winds. Annually, wind speeds are strongest in the WS basic near-surface stability
regime (9.7 m s$^{-1}$), weaker and similar between NN (7.2 m s$^{-1}$) and MS (6.5 m s$^{-1}$) regimes, and then
weakest and similar between the VSM-WSA, SS, and VSS (4.4 m s$^{-1}$ to 5.4 m s$^{-1}$) regimes. A similar
pattern is observed in the fall. However, in the winter NN has slightly stronger winds than WS, and in
summer and spring, wind speeds in the NN and VSM-WSA regimes are similar. Like what was noted at
all the sites above as well, winds in the VSM-WSA regime are 31% to 43% weaker than those in the NN
and WS regimes on an annual basis and in the fall and winter. Wind speed in the VSM-WSA regime is
more like that in the NN regime in the summer and spring, but still over 45% weaker than those in the
WS regime. When considering WS and stronger stability regimes the wind speed generally decreases with
increasing stability. This can be seen in the fall and spring where wind speeds decrease, from WS (9.5 m
s$^{-1}$ to 9.7 m s$^{-1}$) to SS (5.5 m s$^{-1}$), and from WS (10.5 m s$^{-1}$) to VSS (4.4 m s$^{-1}$) in the winter. However, in
the summer, while wind speeds decrease from WS (8.3 m s$^{-1}$) to MS (6.9 m s$^{-1}$), winds then increase to SS
(8.3 m s$^{-1}$).

As stability aloft increases in each stability grouping, wind speed decreases with increasing stability on an annual basis and usually in the winter (Figure 12; Table S5). Wind speed also decreases with increasing stability aloft in the WS and MS stability groupings in the fall and spring, and in the WS group in the summer. This tendency for wind speed to decrease with increasing stability aloft is generally opposite what was observed at the other sites discussed previously (Figures 4, 6, 8, and 10). This decrease in wind speed with increasing stability aloft is usually less than 3 m s$^{-1}$, except in the winter in the NN stability group (5.3 m s$^{-1}$) and in the winter and spring in the WS stability group (both decrease 4.1 m s$^{-1}$). In the summer, fall, and spring as stability aloft increases wind speeds do not differ much in the NN and VSM stability groupings. For example, wind speeds across the NN and VSM regimes in summer differ only by 1.3 m s$^{-1}$ to 1.4 m s$^{-1}$, and in fall, wind speed in the NN regime differ by less than 1 m s$^{-1}$.

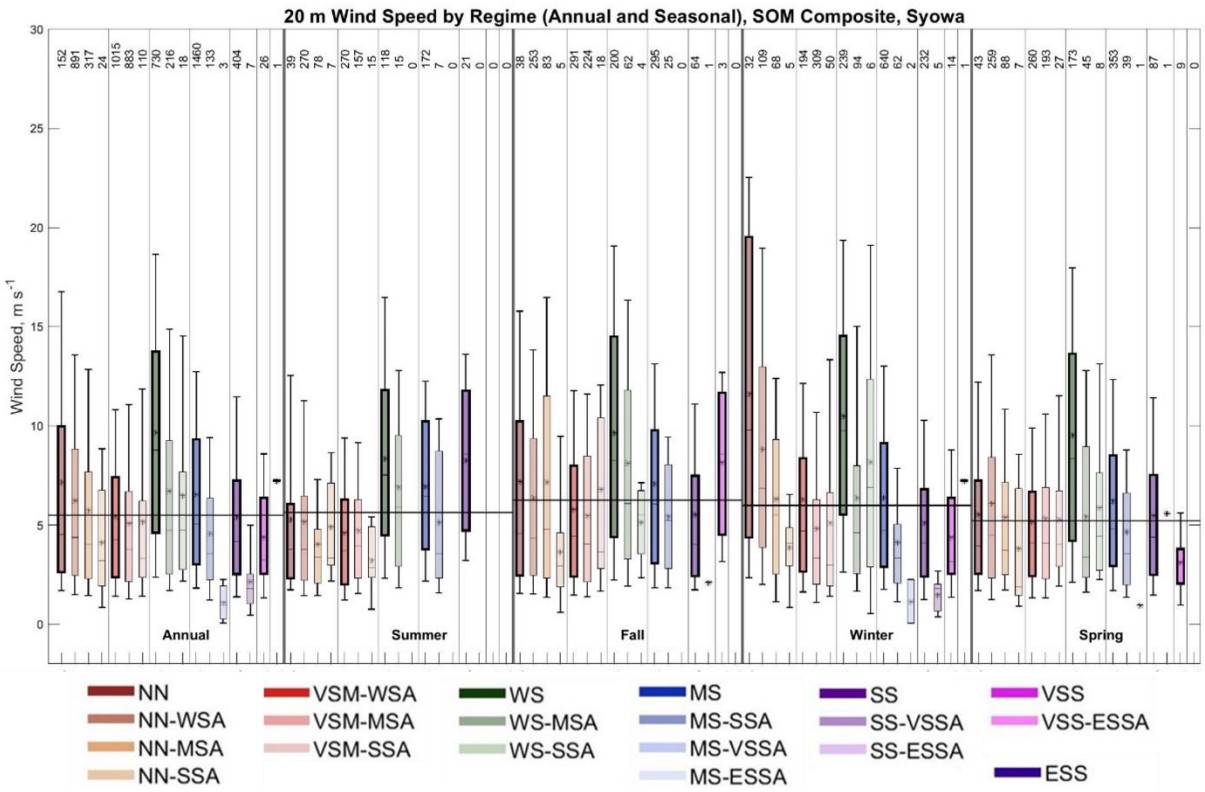

*Figure 12: Box plot showing the distribution of 20 m wind speed observed for each stability regime at Syowa annually (left panel) and seasonally (right four panels – summer, fall, winter, and spring). Box plots show median 20 m wind speed (horizontal line), mean 20 m wind speed (center black star), 25th and 75th percentiles (edges of boxes), and 10th and 90th percentiles (whiskers). The thin vertical black lines in the figure separate the stability groupings in each panel (annual or seasonal). The thin horizontal black lines across each panel (annual or seasonal), indicate the mean value for that entire time period. The numbers at the top indicate the number of radiosonde profiles in each regime.*

**4 Discussion and Conclusions**

To compare and synthesize the forcing mechanisms for varying boundary layer stability across the Antarctic continent from the individual sites presented in the previous section, Figure 13 shows the mean downwelling longwave radiation (left column) and 20 m wind speed (right column) for each stability grouping annually (panels a and b) and seasonally (panels c to j). Here, stability groupings are all stability regimes with the same near surface stability classification regardless of the aloft stability. For example, the mean forcing for the NN stability grouping would include all the NN regimes, regardless of

aloft stability. To further simplify the results shown in this summary figure, any stability grouping which exhibits less than 10 observations total in each season has been omitted from this figure. For example, there is only one ESS observation at Neumayer and Syowa, both in the winter, so these stability groupings are not shown since the mean is likely not very representative.

Figure 13 (left column) shows downwelling longwave radiation generally decreases annually and seasonally with increasing stability from the NN to ESS stability groups, consistent with the results shown in Section 3 for all five study sites. Downwelling longwave radiation usually decreases from NN to VSM, and then slightly increases from VSM to WS. From WS to the strongest stability regime present at a given site in each season, downwelling longwave radiation then usually decreases, except in the summer at the continental interior sites where downwelling longwave radiation is similar across these regimes. Similar to downwelling longwave radiation, downwelling shortwave radiation is also found to generally decrease with increasing stability annually and in the transition seasons (Figure S6). While the magnitude of the change in downwelling shortwave radiation is large (usually >100 W m$^{-2}$) across the range of stability regimes observed in a given season it is important to remember that the high albedo of the Antarctic ice sheet will mute the impact of this large change in downwelling shortwave radiation on the surface energy budget making the forcing from changes in downwelling longwave and downwelling shortwave radiation comparable in their net effect on the surface energy budget. In the summer there generally is not a trend in downwelling shortwave radiation with varying stability (Figure S6).

At Dome C, solar radiation has previously been described as a dominant forcing mechanism, rather than downwelling longwave radiation, in driving changes in stability during this season, unlike in the winter and transition seasons when changes in downwelling longwave radiation are more able to quickly alter near-surface stability (Zhang et al., 2011; Pietroni et al., 2013). However, upon examination of downwelling shortwave radiation at Dome C in the summer (Figure S2), a clear difference in solar radiation was not observed across the regimes. This is likely because the radiosondes at Dome C are launched at approximately 0400 local time, and thus are likely reflective of early morning conditions, namely shallower boundary layers with stronger stability (Dice et al., 2023) after a period of low solar radiation. Further investigation of the forcing mechanisms for variations in boundary layer stability at Dome C in the summer would require higher temporal resolution radiosonde data.

For the 20 m wind speed (Figure 13, right column), considering the first three stability regimes (NN, VSM, and WS), wind speeds are usually strongest in the WS regime, except at Neumayer, while wind speeds are more moderate in NN, and weakest in VSM. This is seen annually and seasonally and highlights an important difference in forcing for the VSM regime in comparison to the NN and WS regimes, from which VSM is derived, having the same potential temperature gradient as these regimes, but with a much shallower boundary layer (Table 2). The relatively weaker winds in VSM in comparison to NN and WS, which was also observed at all sites in Section 3, suggests there is less mechanical generation of turbulence in this regime which results in a shallower boundary layer. At all sites except Dome C, from WS to the strongest stability regime present in each season at a given site, the 20 m wind speed usually decreases. The few exceptions to this behavior are at Neumayer, from SS to VSS annually and in the winter, and from MS to SS in the summer. The increase in 20 m wind speed as stability increases at Dome C is an unexpected result, as previous studies have shown that lower wind speeds are usually associated with stronger stability (Hudson and Brandt, 2005; Cassano et al., 2016; Dice and Cassano, 2022). A discussion as to why this behavior is observed will be given below.

Considering the combined effects of radiative forcing and mechanical mixing on boundary layer stability for the NN, VSM and WS regimes we note unique forcing for each stability grouping. For the NN regime, larger downwelling longwave radiation than in the VSM and WS groups results in reduced

surface cooling or possibly radiative heating, resulting in reduced near-surface stability. Higher
downwelling shortwave radiation (Figure S6) in the NN regime in comparison to the VSM and WS
regimes in most cases in the fall and spring also likely contribute to increased surface heating and the
near-neutral conditions. Also, the winds in NN, which are usually more moderate in comparison to those
in WS, also favor the near-neutral stability of this regime (Cassano et al., 2016, Nigro et al., 2017). The
WS regime usually has lower downwelling longwave radiation in comparison to the NN regime, which
favors slightly enhanced stability in comparison. The stronger winds in the WS regime, compared to NN
and VSM, prevent stability from being any stronger in the WS regime. The VSM regime has distinct
radiative and mechanical forcing compared to the NN and WS regimes. The VSM regime has mean
downwelling longwave radiation between that in the NN and WS regimes favoring stability that is
intermediate to these two regimes. The weaker winds in VSM compared to NN and WS result in less
mechanical generation of turbulence and a shallower boundary layer, which distinguishes this regime
from the NN and WS regimes.
In comparison, for the WS and stronger stability regimes it appears that the decrease in
downwelling longwave radiation, with increasing stability, is the primary forcing that leads to greater near
surface stability, in combination with a general decrease in downwelling shortwave radiation as well
(Figure S6). For all sites, except Dome C, wind speed generally decreases with increasing stability which
also favors stronger near surface stability due to reduced mechanical mixing. The anomalous results at
Dome C will be discussed further below.

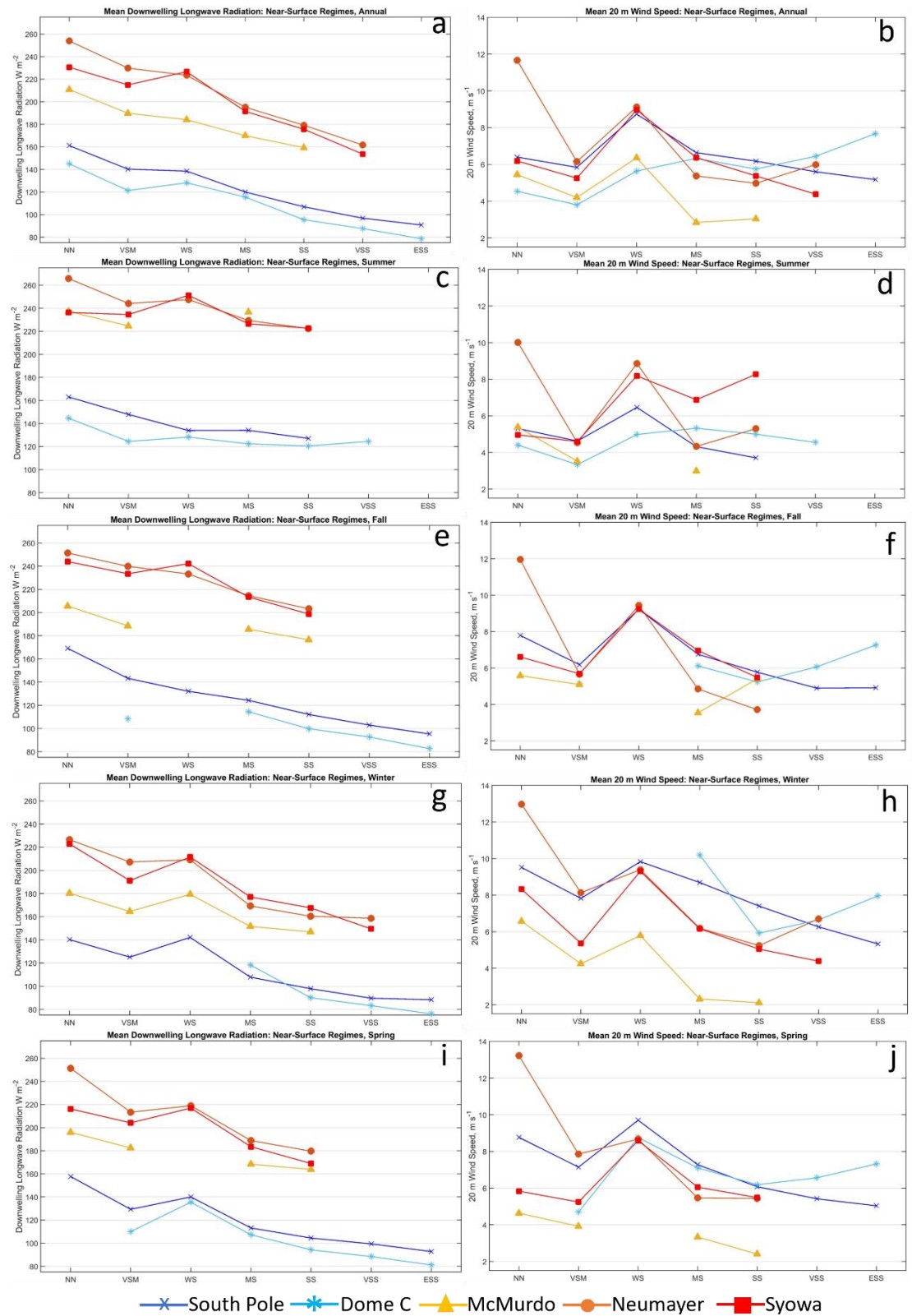

Figure 13: Summary of the mean downwelling longwave radiation (left column) and 20 m wind speed (right column) for the near-surface stability regimes at all five sites annually (a and b) and seasonally: summer, fall, winter, and spring (c through j).

Next, we compare the forcing mechanisms for the basic near-surface stability regimes, with no
enhanced stability aloft, and regimes with the same near surface stability but enhanced stability aloft.
Figure 14 shows the differences in mean downwelling longwave radiation (left column) and 20 m wind
speed (right column) between each basic near-surface stability regime and those regimes with the same
near-surface stability but enhanced stability aloft. The difference is calculated as the mean downwelling
longwave radiation across, for example, NN-WSA, NN-MSA, and NN-SSA minus the mean
downwelling longwave radiation in NN. The magnitude (either positive or negative) of the bar indicates
this difference annually (a and b) and seasonally (c through j) for each site. As with Figure 13, any basic
near-surface stability regime or aloft groupings with less than 10 observations in has been omitted from
this figure and marked with an X. Where differences between the basic near-surface stability regime and
aloft groupings are not statistically significant, the bar has been dulled in color by the addition of white
shading.
The left column in Figure 14 shows that downwelling longwave radiation is almost always lower
for regimes with enhanced stability aloft compared to their basic near-surface stability regime
counterparts, indicated by the consistently negative bars annually and seasonally. Additionally, very few
of these bars are distinguished as not statistically significant, indicating that the differences between the
basic near-surface stability regimes and those with enhanced stability aloft are physically important
differences. These differences mostly range from a few to 15 or more W m$^{-2}$. The magnitude of this
negative difference when enhanced stability is present aloft is usually larger at South Pole compared to
Dome C, which usually has the smallest (or about the same) difference compared to the other sites. Large
differences also occur at Neumayer in the summer for the VSM stability grouping (difference of about 22
W m$^{-2}$), and the spring in the MS stability grouping (difference of about 25 W m$^{-2}$). The largest
differences generally occur at Syowa, especially in the winter where this difference reaches nearly 40 W
m$^{-2}$ in the WS stability grouping.
The right column in Figure 14 shows that 20 m wind speed is almost always higher (indicated by
few bars with white shading) for regimes with enhanced stability aloft compared to the basic near-surface
stability regimes, except at Syowa, with differences typically ranging from less than 0.5 m s$^{-1}$ to about 2
m s$^{-1}$. The magnitude of this difference is usually larger at Dome C (usually between 1 m s$^{-1}$ and 5 m s$^{-1}$)
compared to at South Pole (usually less than 2 m s$^{-1}$), especially when stability is MS and greater. In the
summer, wind speed does not differ as much between the basic and aloft regimes compared to the
difference in the other seasons. In the summer, the smaller difference in wind speed between the basic and
aloft stability regimes in comparison to in the other seasons, suggest that changes in wind speed are not as
important in forcing changes in stability, but rather, like what was noted above, changes in shortwave
radiation contribute more to changes in near-surface stability (Zhang et al., 2011; Pietroni et al., 2013).
Unlike at the other sites, at Syowa (red bars) wind speeds are always less when enhanced stability aloft is
present, and the magnitude of this decrease is usually as large or larger (1 m s$^{-1}$ to 4 m s$^{-1}$) than the
increases in wind speed seen at the other sites.
Considering both the radiative and mechanical forcing differences when enhanced stability aloft
is present provides insights into the mechanisms that result in stability regimes with stronger stability
above the boundary layer. The reduced downwelling longwave radiation when there is enhanced stability
aloft (Figure 14) would suggest that near-surface stability should be stronger, like what was seen in
Figure 13, but instead stability near the surface remains the same with enhanced stability aloft. It is
possible that enhanced near-surface wind-driven mixing could be associated with the passing of synoptic
cyclones, other weather systems, or low-level advection, all of which could increase wind speeds and
decrease stability near the surface leaving behind enhanced stability aloft. To further investigate this
possibility would, however, require higher temporal resolution radiosonde observations.
Without these higher resolution data to validate this, it is hypothesized that the stronger near
surface stability suggested by the reduced downwelling longwave radiation is unable to form due to the
stronger wind and associated mechanical mixing resulting in a layered stability profile, with weaker
stability near the surface and enhanced stability aloft. This suggested behavior is consistent with previous
research that found that as wind speed increases near surface stability is reduced (Hudson and Brandt
2005; Pietroni et al., 2013; Cassano et al., 2016; Silva et al., 2022). The exception is the anomalous
behavior at Syowa, where wind speed is lower for regimes with enhanced stability aloft in comparison to
the basic near-surface stability regimes, and this will be discussed in more detail below.

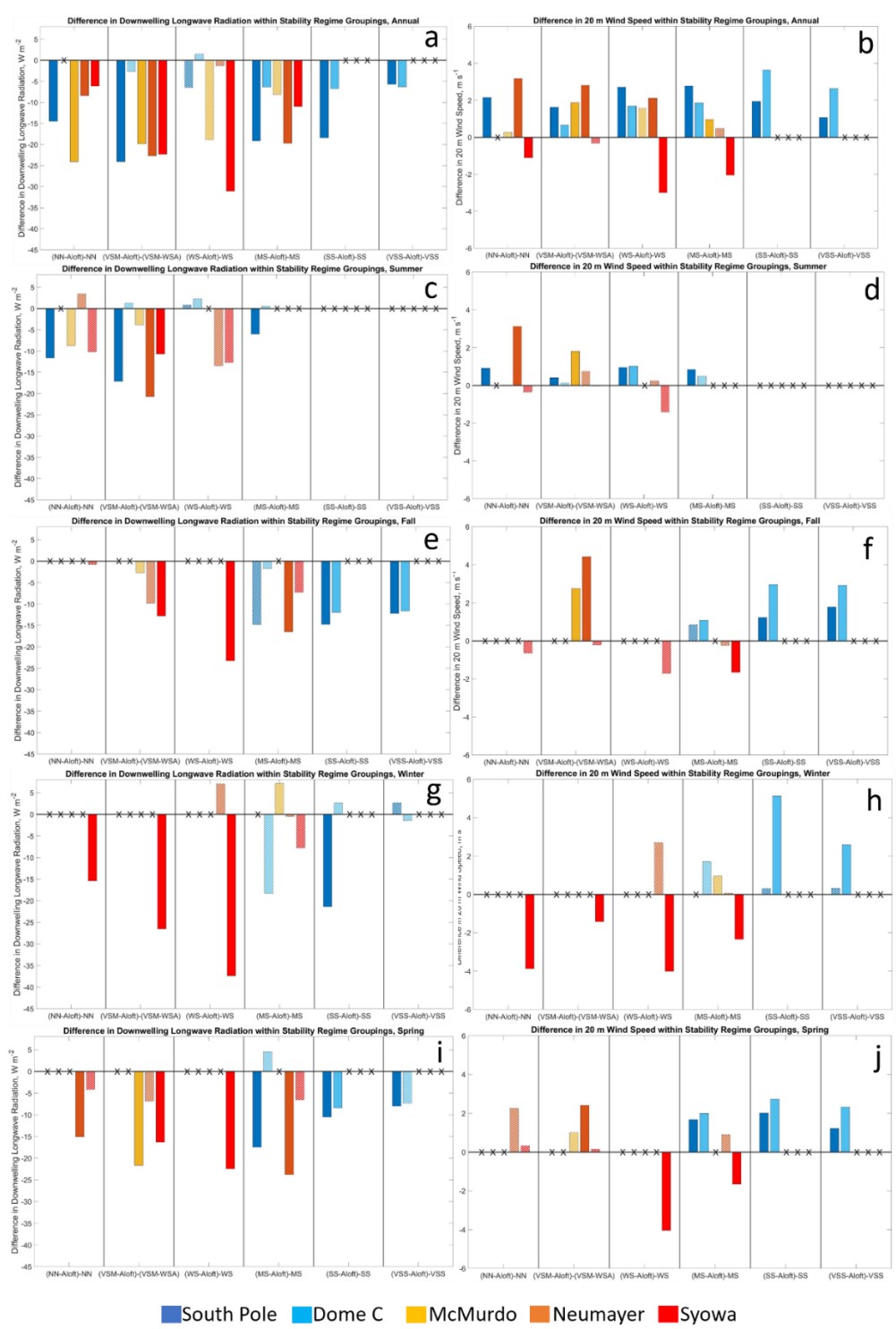

*Figure 14: Summary of the difference in downwelling longwave radiation between the near-surface*
*stability regimes and the mean of the aloft regimes (left column) and the same for 20 m wind speed (right*
*column) at all five sites annually (a and b) and seasonally: summer, fall, winter and spring (c through j).*
*An 'X' in place of a bar indicates fewer than 10 observations are present for either the basic or aloft*
*variations in this regime and has not been included. White shading over a given bar indicates that the*
*difference in downwelling longwave radiation at that site between the aloft regimes and the basic regimes*
*is not statistically significant.*
The results discussed above confirm many of the expectations outlined in the introduction, that
downwelling longwave radiation decreases with increasing stability as does 20 m wind speed for regimes
WS and stronger (Figure 13). The specific forcing for the VSM regime was discovered to be slightly less
downwelling longwave radiation and weaker winds in comparison to the NN and WS regimes which
result in similar stability but less vertical mixing, and a shallower boundary layer. Figure 14 showed that
enhanced near-surface winds counteract the reduced downwelling longwave radiation when enhanced
stability aloft is present, allowing weaker near-surface stability to persist while enhanced stability is
present aloft. There were also some unexpected results, namely the increase in wind speed with increasing
stability at Dome C (Figure 13, right column), and the lower wind speeds with enhanced stability aloft
compared to the basic near-surface stability regimes at Syowa (Figure 14, right column). These
anomalous findings will now be further discussed.
At Dome C, a strong decrease in downwelling longwave radiation with increasing stability in the
winter, fall, and spring is likely responsible for driving changes in stability during these seasons (Figure 5
and Figure 13). In the summer, while other studies have observed changes in solar radiation to be a
driving force of changes in near-surface stability (Zhang et al., 2011; Pietroni et al., 2013), fairly
consistent downwelling shortwave radiation across regimes (Figure S6) was observed at Dome C in the
summer, however this may be due to the timing of the early-morning radiosonde launches at this site.
Stone and Kahl (1991) found surface warming and reduced stability with enhanced downwelling
longwave radiation, and that variations in downwelling longwave radiation are responsible for most of the
variations in changing surface conditions and stability at the South Pole. This is also consistent with the
observations here from the continental interior, particularly at Dome C Additionally, Pietroni et al. (2013)
found changes in stability in the winter at Dome C to be mostly attributed to sudden increases
downwelling longwave radiation. The unexpected result at Dome C is that wind speed increases with
increasing stability, counter to previous results (Hudson and Brandt 2005; Pietroni et al., 2013; Cassano et
al., 2016; Silva et al., 2022).
It is hypothesized that the stronger wind speed with increasing stability is not contributing to the
formation of the stability regimes, but rather that the increase in wind speed is a response to the greater
stability. In these stronger stability regimes, turbulence generated by wind-driven mixing is suppressed by
increasingly strong buoyancy forces, resulting in a complicated relationship between wind speed and
stability. Specifically, when stability is strong, the boundary layer can become mechanically decoupled
from the surface (Banta et al., 2007; Vignon et al., 2017). The very low values of downwelling longwave
radiation at Dome C led to strong surface cooling and the development of strong stability, especially
immediately adjacent to the ice surface, which resulted in weak or intermittent turbulence (Pietroni et al.,
2013; Zhang et al., 2011). With little turbulence, frictional slowing of the wind will be reduced, and 20 m
winds could increase with increasing stability. The reason this behavior occurs at Dome C, but is not
observed at the other sites, is unclear. It may be due to the very strong radiative cooling at this highest
elevation site considered in this study. Also, unlike the other sites, Dome C is almost flat so no katabatic
flow can develop to advect away the radiatively cooled air adjacent to the surface, allowing strong
stability to grow with time while turbulence is suppressed.
At Syowa, unlike at the other sites, wind speed was less when enhanced stability aloft was present
and does not follow the conclusion that increased wind speed is responsible for reducing near surface
stability (Figure 14). This leaves to question the forcing mechanism for regimes with enhanced stability
above a layer of weaker near-surface stability at Syowa. We suggest that the answer is likely related to the
complex katabatic and cyclonic influences that are present at Syowa and have been shown to impact the
boundary layer conditions at this site (Murakoshi, 1958; Tomikawa et al., 2015; Yamada and Hirasawa,
2018). At Syowa, easterly winds are associated with windy, cyclonic activity and weak near-surface
stability, while southerly or southwesterly winds are associated with calm, non-cyclonic conditions and
moderate to strong stability (Tomikawa et al., 2015; Yamada and Hirasawa, 2018). Supplemental Figure 7
provides some insight for this by showing the range of wind direction observed for each stability regime
annually and seasonally at Syowa. As stability aloft increases in each stability grouping, the wind
direction changes from easterly to more southeasterly. As the wind direction shifts from easterly to
southeasterly the wind has a more continental origin and is likely colder. This suggests that weak drainage
flow from the continental interior may be advecting cold air at low levels, while more mild, maritime air
remains aloft, resulting in profiles with enhanced stability aloft at the interface between the cold
continental air at low levels and the mild maritime air above.
Here, the forcing mechanisms for the variations in boundary layer stability described by Dice et
al. (2023) were identified for two continental interior sites and three coastal sites in Antarctica. Boundary
layer stability and the forcing mechanisms that drive variations in boundary layer stability is widely
misrepresented in weather and climate models (e.g., Genthon et al., 2013; Holtslag et al., 2013; Mahrt,
2014). A next step in this work will be to assess the ability of the Antarctic Mesoscale Prediction System
(AMPS) (Powers et al., 2012) to simulate the frequency of boundary layer stability regimes (Dice et al.,
2023) and differing forcing for each stability regime.

**Data Availability**

The data used to support this project can be found at:
McMurdo:
All data: https://adc.arm.gov/discovery/#/results/site_code::awr.
Syowa:
Radiosonde data: Office of Antarctic Observation Japan Meteorological Agency (pers. comm.
Yutaka Ogawa)
Radiation data: https://doi.pangaea.de/10.1594/PANGAEA.956748
Dome C:
Radiosonde data: https://www.climantartide.it/dataaccess/index.php?lang=it
Radiation data: https://doi.pangaea.de/10.1594/PANGAEA.935421
South Pole:
Radiosonde data: http://amrc.ssec.wisc.edu/data/ftp/pub/southpole/radiosonde/
Radiation data: https://doi.pangaea.de/10.1594/PANGAEA.956847
Neumayer:
Radiosonde data: https://doi.org/10.1594/PANGAEA.940584
Radiation data: https://doi.org/10.1594/PANGAEA.940584

**Competing Interests**

The contact author has declared that none of the authors has any competing interests.
**Acknowledgements**
Funding for this work came from the United States National Science Foundation (NSF) grant OPP
1745097 and the National Aeronautics and Space Administration (NASA; award 80NSSC19M0194).The
authors thank the United States Antarctic Program, the Department of Energy, the Baseline Surface
Radiation Network, the Antarctic Meteorological Research and Data Center, the Antarctic Meteo-
Climatological Observatory, and the Office of Antarctic Observation Japan Meteorological Agency for
the support and logistics for the data used in this paper.

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
