# Peer review of "Forcing For Varying Boundary Layer Stability Across Antarctica"

_EGUsphere, 2023_

## Referee Comment (RC1)

**Review of "Forcing For Varying Boundary Layer Stability Across Antarctica", by Dice et al. (egusphere-2023-2062)**

**General**

This paper follows up findings presented in an earlier study (egusphere-2023-1673) in which the climatology of boundary-layer and lower atmosphere stability was examined at five Antarctic stations. In that paper, a system was developed to classify stability regimes based on potential temperature gradients within and just above the boundary layer. In this follow-up paper, the variations of two possible forcing factors – downwelling longwave radiation and near-surface wind speed – across the stability regimes are examined at the five stations and conclusions are drawn about the roles of these two factors in driving the regimes.

The paper is a logical follow-on from egusphere-2023-1673 and does provide some useful insight into the controls on near-surface atmospheric stability across Antarctica. It is generally well-written, although with 20 different stability regimes being examined at five stations during four seasons it can get quite hard to follow at times. The figures are informative but may not all be clearly readable at publication size – the legibility of some text is marginal even in the preprint version.

I do have some concerns about the methodology and approach used in the study, which I have set out below. Adequately addressing these points may require fairly major revision, but once this has been done the paper should be suitable for publication in *WCD*.

**Major points**

1. The paper examines variations in stability associated with two forcing factors – near-surface wind speed and downwelling longwave radiation. The reasoning behind this is that, within the boundary layer, the potential temperature gradient results from an interplay between the surface heat flux and mixing by turbulence. Over the Antarctic ice sheets there is a reasonably close relationship between the surface heat flux and the net radiative flux at the surface. Hence, outside the summer season, downwelling longwave should be a reasonable proxy for surface heat flux. During summer, shortwave radiation is a significant contributor to the surface energy balance and needs to be taken into account. This is mentioned (lines 660-665), but hasn't been followed up – surely shortwave data are available for all stations? Windspeed is probably a reasonable proxy for mixing in the near neutral to moderately stable regimes, but things get more complicated at higher stabilities as buoyancy forces increasingly suppress the mechanical production of turbulence. The relationship between wind speed and mixing is, therefore, not straightforward in this high stability range (see Vignon et al, 2017, referenced in the manuscript). This may explain some of the counterintuitive behaviour seen in, e.g., fig. 13.

2. In much of your analysis I think that you are making an implicit assumption that the boundary layer is in equilibrium with its forcing by radiative cooling and wind-driven mixing. This may not always be the case, particularly at the coastal sites where conditions can change rapidly as a result of the movement of synoptic-scale cyclones and other weather systems. Your VSML regime, for example, could be the result of a sudden increase in wind speed following an extended period of surface cooling resulting in the erosion of the surface inversion from below by wind-driven mixing. Your data don't have sufficient temporal resolution to study this in detail but I do think that you should mention it as a possible limitation to your analysis.

3. In your analysis, you study variations in stability both within the boundary layer and in the layer between the top of the boundary layer and the upper limit of your study (500m above surface level). By your definition, there is no (or very limited) turbulent mixing within this upper layer and it is largely decoupled from the surface (and hence from the surface energy balance). I would thus argue that it is not appropriate to try to explain variations in stability in this upper layer in terms of wind speed and downwelling longwave radiation as these parameters are strongly associated with the structure of the turbulent boundary layer but will not directly influence stability in the free atmosphere above. Stability in this upper layer will be influenced by things such as advection, subsidence, moist diabatic processes and radiative flux divergence. The thermal structure of this layer could also be a "relic" of a deep, stable boundary layer that has been modified at low levels as a result of increased wind-driven mixing (see also my point 1 above) or reduced surface heat flux due to an increase in downwelling longwave radiation. You could discuss these points further, e.g. in the paragraph starting at line 734.

**Minor points**

1. L180-181: The sentence starting "Above 20m…" is a bit confusing and needs rephrasing, e.g. "The 20 m lower limit was chosen because radiosonde measurements below this level are often biased warm". You have already mentioned (L. 156) that you do not use data below 20m because the sonde may not be in equilibrium.

2. Figure 2: Are all of these profiles from the same station? Are they single profiles or means for the class? What is the temperature anomaly with respect to?

3. L 508-509: Sentence starting "In the winter…" is incomplete – it does not have a verb.

4. L665: Delete "during the summer months" to aboid repetition.

5. L685: "lower", not "less".

6. Figure 13: Presentational point – the horizontal axis does not show a continuous variable so, strictly, you should not join the points with a line.

7. L 712: "…is almost always lower…". Is the difference statistically significant?

---

## Author Comment (AC1)

**Response to anonymous referee comments**

The authors thank the two referees for taking the time to review this manuscript and for their helpful comments, which have improved the manuscript. The text in the updated manuscript reflecting the changes made in this document is included below each response in quotation marks. Responses to referee comments are in *italics*. In areas where the response to the referee includes a specific edit or addition to text in the manuscript, these edits are noted in *highlighted yellow*. *Line numbers referencing a change are those in the clean revised manuscript.*

**Reviewer 1:**

Review of "Forcing For Varying Boundary Layer Stability Across Antarctica", by Dice et al.

**General**

This paper follows up findings presented in an earlier study (egusphere-2023-1673) in which the climatology of boundary-layer and lower atmosphere stability was examined at five Antarctic stations. In that paper, a system was developed to classify stability regimes based on potential temperature gradients within and just above the boundary layer. In this follow-up paper, the variations of two possible forcing factors – downwelling longwave radiation and near-surface wind speed – across the stability regimes are examined at the five stations and conclusions are drawn about the roles of these two factors in driving the regimes.
The paper is a logical follow-on from egusphere-2023-1673 and does provide some useful insight into the controls on near-surface atmospheric stability across Antarctica. It is generally well-written, although with 20 different stability regimes being examined at five stations during four seasons it can get quite hard to follow at times. The figures are informative but may not all be clearly readable at publication size – the legibility of some text is marginal even in the preprint version.
I do have some concerns about the methodology and approach used in the study, which I have set out below. Adequately addressing these points may require fairly major revision, but once this has been done the paper should be suitable for publication in WCD.

*Thank you for the thoughtful comments, we look forward to addressing these concerns below.*

**Major points**

1. The paper examines variations in stability associated with two forcing factors – near-surface wind speed and downwelling longwave radiation. The reasoning behind this is that, within the boundary layer, the potential temperature gradient results from an interplay between the surface heat flux and mixing by turbulence. Over the Antarctic ice sheets there is a reasonably close relationship between the surface heat flux and the net radiative flux at the surface. Hence, outside the summer season, downwelling longwave should be a reasonable proxy for surface heat flux. During summer, shortwave radiation is a significant contributor to the surface energy balance and needs to be taken into account. This is mentioned (lines 660-665), but hasn't been followed up – surely shortwave data are available for all stations? Windspeed is probably a reasonable proxy for mixing in the near neutral to moderately stable regimes, but things get more complicated at higher stabilities as buoyancy forces increasingly suppress the mechanical production of turbulence. The relationship between wind speed and mixing is, therefore, not straightforward in this high stability range (see

Vignon et al, 2017, referenced in the manuscript). This may explain some of the counterintuitive behaviour seen in, e.g., fig. 13.

*Thank you for this comment as your thoughts regarding the importance of shortwave radiation and the interplay between wind and stability for the stronger stability regimes are quite useful. Shortwave radiation is an important variable to further explain and show when discussing these results. Shortwave data is available for all stations. For this reason, Figures S1-S6 have been added to the supplemental information. Figures S1-S6 show, similar to the box plots shown throughout the main text of the manuscript, boxplots depicting the range of downwelling shortwave radiation for each site and each stability regime annually and seasonally. The following text has been added in the results sections to further explore this variable:*

*In lines 294-308 in the South Pole results section:*

[revised manuscript text omitted]

Regarding the more complicated relationship between wind-driven mixing and stability in the higher stability regimes, this is further clarified in lines 868-871 when discussing this complicated relationship at Dome C in the discussion section:

"In these stronger stability regimes, turbulence generated by wind-driven mixing is suppressed by increasingly strong buoyancy forces, resulting in a complicated relationship between wind speed and stability. Specifically, when stability is strong, the boundary layer can become mechanically decoupled from the surface (Banta et al., 2007; Vignon et al., 2017)."

2. In much of your analysis I think that you are making an implicit assumption that the boundary layer is in equilibrium with its forcing by radiative cooling and wind-driven mixing. This may not always be the case, particularly at the coastal sites where conditions can change rapidly as a result of the movement of synoptic-scale cyclones and other weather systems. Your VSML regime, for example, could be the result of a sudden increase in wind speed following an extended period of surface cooling resulting in the erosion of the surface inversion from below by wind-driven mixing. Your data don't have sufficient temporal resolution to study this in detail but I do think that you should mention it as a possible limitation to your analysis.

*Thank you for this comment. The following text has been included in the beginning of the Results section to reflect these additional important factors in shaping boundary layer stability in lines 255-258:*

*"In addition to these two variables, additional forcing mechanisms, such as the passing of synoptic cyclones or other weather systems, or low-level advection, all of which could result in changes in near surface stability, are possible, although not investigated at length in this analysis."*

*The following text has been added to lines 820-826 in the discussion section to reflect this potential limitation:*

*"Considering both the radiative and mechanical forcing differences when enhanced stability aloft is present provides insights into the mechanisms that results in stability regimes with stronger stability above the boundary layer. The reduced downwelling longwave radiation when there is enhanced stability aloft (Figure 14) would suggest that near-surface stability should be stronger, like what was seen in Figure 13, but instead stability near the surface remains the same with enhanced stability aloft. It is possible that enhanced near-surface wind-driven mixing could be associated with the passing of synoptic cyclones, other weather systems, or low-level advection, all of which could increase wind speeds and decrease stability near the surface leaving behind enhanced stability aloft. To further investigate this possibility would, however, require higher temporal resolution radiosonde observations.*

*Without these higher resolution data to validate this, it is hypothesized that the stronger near…"*

3. In your analysis, you study variations in stability both within the boundary layer and in the layer between the top of the boundary layer and the upper limit of your study (500m above surface level). By your definition, there is no (or very limited) turbulent mixing within this upper layer and it is largely decoupled from the surface (and hence from the surface energy balance). I would thus argue that it is not appropriate to try to explain variations in stability in this upper layer in terms of wind speed and downwelling longwave radiation as these parameters are strongly associated with the structure of the turbulent boundary layer but will not directly influence stability in the free atmosphere above. Stability in this upper layer will be influenced by things such as advection, subsidence, moist diabatic processes and radiative flux divergence. The thermal structure of this layer could also be a "relic" of a deep, stable boundary layer that has been modified at low levels as a result of increased wind-driven mixing (see also my point 1 above) or reduced surface heat flux due to an increase in downwelling longwave radiation. You could discuss these points further, e.g. in

the paragraph starting at line 734.

*The purpose of explaining and defining the stability above the boundary layer is, as you mentioned, to describe how increased wind-driven mixing or radiative forcing near the surface may result in boundary layers with reduced near surface stability and enhanced stability "aloft" (e.g., above the boundary layer). Additionally, understanding what the stability is like immediately above the boundary layer is important for the evolution of the boundary layer in time, as enhanced stability aloft will affect how the boundary layer grows. This intention is further clarified in the methods section with the following text in lines 186-190:*

*"Stability regimes aloft, ==just above the boundary layer,== were also defined, as many of the radiosonde profiles have enhanced stability above layers of weaker, near-surface stability. ==It is important to identify the stability structure both within, and just above the boundary layer for understanding of its evolution in time.== ==For example, enhanced stability above the boundary layer could act to suppress the growth of the boundary layer with strong radiative forcing or mechanical mixing."==*

*We also note that there is a consistent signal seen in downwelling longwave radiation and 20 m wind speed between regimes which have enhanced stability aloft and those which do not, and thus these important differences are described in the text. There may be some degree of "memory" or a "relic" of the atmosphere above the boundary layer when enhanced stability exists aloft and downwelling longwave radiation is lower for these cases (which as mentioned in the text) would support strong stability, in comparison to the higher downwelling longwave radiation in the regimes without enhanced stability aloft. This has been incorporated into the new text addressing comment #2 above.*

*Additionally, the word "surface" in line 818 has been replaced by "boundary layer" to further clarify this.*

**Minor points**

1. L180-181: The sentence starting "Above 20m…" is a bit confusing and needs rephrasing, e.g. "The 20 m lower limit was chosen because radiosonde measurements below this level are often biased warm". You have already mentioned (L. 156) that you do not use data below 20m because the sonde may not be in equilibrium.

*This sentence has been removed to avoid redundancy.*

2. Figure 2: Are all of these profiles from the same station? Are they single profiles or means for the class? What is the temperature anomaly with respect to?

*These profiles are from the same station. They are the mean profiles for each stability regime observed at Dome C, and are just meant to show what the profiles of the stability regimes look like. The potential temperature anomaly is with respect to the 20 m potential temperature. The following text is added in the figure caption to clarify this (highlighted yellow):*

*"Figure 2: Examples of the vertical profile structure of the regimes listed in Table 3. The potential temperature gradient is shown in pink (top axis), the potential temperature anomaly ==(with respect to the==*

*20 m potential temperature form the radiosonde)* *is shown in blue (bottom axis). The stability regime acronym is given above the top left corner of each subplot and is also indicated by the colored outline around each plot, according to the key in the bottom right of the figure."*

3. L 508-509: Sentence starting "In the winter..." is incomplete – it does not have a verb.

*This text has been updated:*

*"In the winter, moderate or strong stability, either near the surface or aloft above a layer of weaker stability,* *is often present* *(85.2%)."*

4. L665: Delete "during the summer months" to avoid repetition.

*This has been deleted.*

5. L685: "lower", not "less".

*This has been changed.*

6. Figure 13: Presentational point – the horizontal axis does not show a continuous variable so, strictly, you should not join the points with a line.

*While we agree with your point that the data shown is not a continuous variable, we believe that connecting these points with lines will help the reader visually follow the trends across stability regimes at each of the individual sites, shown in different colors, so we prefer to maintain the figure in its original form.*

7. L 712: "...is almost always lower...". Is the difference statistically significant?

*Thank you for this comment. We have adjusted Figure 14 to include white shading to indicate instances where differences in downwelling longwave radiation or 20 m wind speed are NOT statistically significant, and the following text has been added to lines 788-790:*

*"Where differences between the basic near-surface stability regime and aloft groupings are not statistically significant, the bar has been dulled in color by the addition of white shading."*

*And in lines 793-796:*

*"Additionally, very few of these bars are distinguished as not statistically significant, indicating that the differences between the basic near-surface stability regimes and those with enhanced stability aloft are physically important differences."*

*And in lines 803-804:*

*"The right column in Figure 14 shows that 20 m wind speed is almost always higher* ==(indicated by few bars with white shading)== *for regimes with enhanced stability aloft compared to the basic near-surface stability regimes…"*

*And to the Figure 14 caption:*

*"Figure 14: Summary of the difference in downwelling longwave radiation between the near-surface stability regimes and the mean of the aloft regimes (left column) and the same for 20 m wind speed (right column) at all five sites annually (a and b) and seasonally: summer, fall, winter and spring (c through j). An 'X' in place of a bar indicates fewer than 10 observations are present for either the basic or aloft variations in this regime and has not been included.* ==White shading over a given bar indicates that the difference in downwelling longwave radiation at that site between the aloft regimes and the basic regimes is not statistically significant."==

**Community comment (Gabin Urbancic):**

**Review of the paper**

"Forcing For Varying Boundary Layer Stability Across Antarctica"

This paper uses an extensive Antarctic dataset to study the forcing mechanisms behind different stability regimes observed and characterized in Dice et al. 2023. This work is of high interest to the community.
I have some issues with the organization of the study which should be addressed with major revisions. The biggest difficulty is organizing the results between statistical predictive and physical results. The terminology and organization make these distinctions difficult and can be misleading. I believe this work can be published after care is taken in improving the terminology and presentation style of the results.

*Thank you for your comments, we look forward to addressing these concerns below.*

**Major Comments:**

1. I have difficulty in understanding the research program for this work. When discussing stability, what is being discussed in the occurrence and properties of the surface based inversion and inversion aloft. Stability, as implemented in parameterizations use the different Richardson numbers or Obukhov stability parameter for the surface layer. In this study the temperature profile is exclusively used to characterize the stability regime.

*For this work we have chosen to define the static stability of the boundary layer by the potential temperature gradient profile. As you point out there are several other ways to define stability including Richardson number and Obukhov length. While these are useful, and used in numerical parameterizations of the boundary layer, we feel the use of vertical potential temperature gradients is also useful. While the Richardson number reflects turbulence intensity, relative to both static stability and mechanically generated turbulence, similar values of Richardson number can arise from a variety of atmospheric states. In contrast, the potential temperature gradient uniquely defines the thermodynamic state of the atmosphere which spans a wide range across the thousands of radiosonde profiles used in*

*this study. Here, the use of the Obukhov length to define stability is not possible since this data set does not include the fluxes necessary to calculate it. We also want to be able to apply this stability analysis to the Arctic (Jozef et al., 2023) and radiosondes are the most easily accessible form of data to accomplish this task.*

The terminology used is also unclear. It is confusing to me to characterize the state of the boundary layer with what is happening outside the boundary layer. The influence of the inversion aloft on the possible states of the boundary layer is of definite interest (as nicely considered in Fig. 14) but is presented in a convoluted way by being directly introduced in one characterization. A simpler format would be defining the states of the boundary layer, and then considering how those states are influenced by the external structure of the 'lower' atmosphere. It is misleading to discuss 'boundary layer stability' when what is considered is the state of the lower atmosphere.

*We appreciate the concern raised by this comment and the reviewer is correct that the aloft stability portion of our classification is outside of the boundary layer. We want to remind the reviewer that we have already published two papers (Dice et al., 2023; Jozef et al., 2023) that use the same boundary layer regime definitions used in this study. As such we do not want to significantly deviate from how these stability regimes were defined or discussed in those published papers.*

*We also note that the stability immediately above the boundary layer is relevant to boundary layer processes. As shown in our results physical insights into boundary layer processes can be gained by considering the stability just above the boundary layer in addition to the stability within the boundary layer. Unique forcing mechanisms can result in reduced near-surface stability (in the boundary layer) while enhanced stability remains aloft (above the boundary layer). This helps to inform how stability in the boundary layer becomes reduced while the potentially original (and stronger) stability remains in a layer above that has been impacted by, for example, increased wind speed or downwelling longwave radiation. Additionally, stability immediately above the boundary layer impacts how boundary layer depth may evolve with time. To make clear why we define boundary layer stability regimes that are based on both static stability within and just above the boundary layer we have added the following text in the methods section in lines 186-190:*

*"Stability regimes aloft, ==just above the boundary layer,== were also defined, as many of the radiosonde profiles have enhanced stability above layers of weaker, near-surface stability. ==Considering the stability both within, and just above the boundary layer is relevant for understanding the time evolution of the boundary layer.=="*

The two forcing mechanisms are downwelling longwave radiation and wind speed (as a proxi for turbulent mixing). The discussion is presented as these forcing mechanisms drive the variations in the boundary layer stability. Again, this is quite complicated. Downwelling LW is due to cloud cover which may (?) reside in the lowest 500 m and therefore can be coupled to the temperature profile and therefore the stability regime. Similarly, the 20 wind speed is highly effected by stability and is an internal variable as understood through Monin-Obukhov Similarity Theory. The presentation in this work discusses the two forcing mechanisms as if they are external parameters, downwelling LW may in fact be but 20 m wind speed certainly is not.

*Thank you for this comment. You are correct that both of the forcing variables we analyze can also be driven by boundary layer stability and as such these forcing variables are not fully independent of the stability regimes we are trying to understand. As you stated, downwelling longwave radiation will respond strongly to cloud cover and some clouds may be within the lowest 500 m of the atmosphere that define our stability regimes, although most clouds will be above this lowest part of the atmosphere making the downwelling longwave radiation more of an independent variable, as you mention. The manuscript already mentions that wind speed can be a function of stability, rather than a variable which alters it. When discussing the anomalous results regarding increasing wind speeds with increasing stability at Dome C, it is mentioned that this is likely that this is a result of the strong stability present and rather is not impacting stability at this site. Additionally, the following text has been added to further clarify the possibility of these variables responding to changes in stability in the results section in lines 252-255:*

*"While downwelling longwave radiation is largely independent of stability, wind speeds can change in response to changes in stability. For example, with very strong near-surface stability, winds can become decoupled from the frictional, slowing effects of the surface and increase."*

If the assumption is that the 20 m wind speed and downwelling LW are the two main drivers of the surface energy balance then it could explain the structure in the atmospheric surface layer assuming that SW radiation is negligible. The rest of the boundary layer and atmosphere above the surface layer are influenced by the time integral of the surface driven fluxes as well as advective, non-local processes. The best that can be done for predicting the 'lower' atmosphere properties is therefore finding statistically representative parameters.

*We agree with the point raised in this comment – that stability above the surface layer reflect many other processes than those being assessed with downwelling longwave radiation and 20 m wind speed. We argue that since meaningful signals are observed in the downwelling longwave radiation and 20 m wind speed observations when enhanced stability aloft is present versus when there is not enhanced stability aloft, there are still useful insights about the processes shaping stability above the surface layer that can be gained from the analysis presented in this manuscript. It is true that the other processes you list in your comment are important for boundary layer stability, they are not further investigated here, as these processes cannot be assessed due to the relatively low temporal resolution of this dataset (radiosonde launches only every 12 or 24 hours). This point has been further clarified in the text in lines 255-258 in the edited (clean) manuscript:*

*"In addition to these two variables, additional forcing mechanisms, such as the passing of synoptic cyclones or other weather systems, or low-level advection, all of which could result in changes in near surface stability, are possible, although not investigated at length in this analysis."*

*And in the discussion section in lines 820-826:*

*"Considering both the radiative and mechanical forcing differences when enhanced stability aloft is present provides insights into the mechanisms that results in stability regimes with stronger stability*

*above the boundary layer. The reduced downwelling longwave radiation when there is enhanced stability aloft (Figure 14) would suggest that near-surface stability should be stronger, like what was seen in Figure 13, but instead stability near the surface remains the same with enhanced stability aloft. ==It is possible that enhanced near-surface wind-driven mixing could be associated with the passing of synoptic cyclones, other weather systems, or low-level advection, all of which could increase wind speeds and decrease stability near the surface leaving behind enhanced stability aloft. To further investigate this possibility would, however, require higher temporal resolution radiosonde observations.==*

> *==Without these higher resolution data to validate this,== it is hypothesized that the stronger near…"*

There is no turbulence dataset that is extensive enough to directly investigate this topic with the generality of this study. Statistical methods are therefore needed to find representative parameters such as downwelling LW and 20 m wind speed, or using temperature profiles to characterize the state of the boundary layer. That being said, I think it is important to make sure the program of how this is treated is made clear.

*The goal of this work is not to identify robust statistical relationships between radiation, wind speed and stability but rather to document how these variables are related. Future work will use the framework developed here to assess whether atmospheric numerical models can replicate the observed relationships between downwelling longwave radiation, 20 m wind speed and boundary layer stability regimes. The results of this future work will provide guidance to the modeling community regarding potential shortcomings in how these processes are represented.*

2. As this is not a "Part 2-.." to the Dice et al. 2023 paper, this work would benefit a higher level of independence. Without reading Dice et al. 2023 the choice of stability regimes seem arbitrary. This work should not repeat all the analysis but should provide information on how the 20 different regimes were chosen.

*A more extensive summary of the regime definition method was originally included in the manuscript in the Data and Methods section. However, the depth of this summary was reduced per a request from the editor before the manuscript went into the preprint discussion period and as such we do not feel that it would be appropriate to expand this discussion against the editor's original recommendation.*

**Minor Comments:**

All Figure – The figures are very dense and often a bit hard to read. Please try to clean up the figures.

*Thank you for the comment. We have removed the regime name labels on the x axis on Figures 3-12 and replaced it with a key to alleviate the density of these figures.*

Fig 13 – I think it is misleading to plot lines in these panels, what does the halfway point between WS and MS mean? The choice of stability bins has no scale and therefore should not be treated as points on an axis. Additionally, one could actually plot the full variables Downwelling LW and U vs dT for the near-surface stability. An improved figure would be for example. U vs dT with bin averaging chosen with scale and then marking the bounds of NN, VSM, ... .

*While we agree with your point that the data shown is not a continuous variable, we believe that connecting these points with lines will help the reader visually follow the trends across stability regimes at each of the individual sites, shown in different colors, so we prefer to maintain the figure in its original form.*

*We appreciate the suggestion for including the potential temperature gradient as the horizontal axis in the figure but choose to define the bins as the stability regimes themselves since their gradients are defined earlier in the methods section. Additionally, this provides a consistent approach of analyzing variables across regimes as has been done in the rest of the manuscript.*

---

## Author Response (AR2)

**Response to anonymous referee comments**

The authors thank the two referees for taking the time to review this manuscript and for their helpful comments, which have improved the manuscript. The text in the updated manuscript reflecting the changes made in this document is included below each response in quotation marks. Responses to referee comments are in *italics*. In areas where the response to the referee includes a specific edit or addition to text in the manuscript, these edits are noted in highlighted yellow. Line numbers referencing a change are those in the clean revised manuscript.

1. Lines 104-105, "Data" section: The paper now includes an analysis of shortwave radiation data, which should be mentioned in the "Data" section.

*This addition has been made in line 105 of the revised manuscript:*

*"Radiosonde data from two continental interior sites (South Pole and Dome C) and three coastal sites (McMurdo, Neumayer, and Syowa) (Figure 1, Table 1) as well as corresponding downwelling longwave and shortwave radiation data at the time the radiosonde launches occurred are included in this analysis."*

2. Table 1: The column "Radiation Instrument and Accuracy" only lists the longwave instrument (pyrgeometer) used at each station. Now that shortwave radiation is also analysed, the shortwave instruments (pyranometers) should also be listed.

*Information regarding the pyranometers for each of the study sites has been added to Table 1.*

3. Figure 2: State in the caption that these profiles are for Dome C.

*The figure caption has been updated:*

*"Figure 2: Examples (from Dome C) of the vertical profile structure of the regimes listed in Table 3. The potential temperature gradient is shown in pink (top axis), the potential temperature anomaly (with respect to the 20 m potential temperature form the radiosonde) is shown in blue (bottom axis). The stability regime acronym is given above the top left corner of each subplot and is also indicated by the colored outline around each plot, according to the key in the bottom right of the figure."*

4. Figure 13: I'm still unhappy about the lines joining the points on this figure - the second referee also raised the same concern. I'm happy to leave it to the Editor to rule on this point.

*The editor has suggested the remove of lines joining the points in Figure 13. The lines have been removed. The same has been done for Figure S6.*